# Dietary regimens appear to possess significant effects on the development of combined antiretroviral therapy (cART)-associated metabolic syndrome

**Boniface M. Chege**[1,2]*, **Peter W. Mwangi**[2], **Charles G. Githinji**[2], **Frederick Bukachi**[2]

**1** School of Health Sciences, Dedan Kimathi University of Technology, Nyeri, Kenya, **2** Department of Human Anatomy and Medical Physiology, University of Nairobi, Nairobi, Kenya

☯ These authors contributed equally to this work.
* bmchege87@gmail.com

**Data Availability Statement:** All relevant data are within the paper and its Supporting information files.

## Abstract

### Introduction

This study investigated the interactions between a low protein high calorie (LPHC) diet and an integrase inhibitor-containing antiretroviral drug regimen (INI-CR)in light of evidence suggesting that the initiation of cART in patients with poor nutritional status is a predictor of mortality independent of immune status.

### Methods

Freshly weaned Sprague Dawley rats (120) were randomized into the standard, LPHC and normal protein high calorie (NPHC) diet groups (n = 40/group) initially for 15 weeks. Thereafter, experimental animals in each diet group were further randomized into four treatment sub-groups (n = 10/group) Control (normal saline), group 1(TDF+3TC+DTG and Tesamorelin), group 2 (TDF+3TC+DTG), and Positive control (AZT+3TC+ATV/r) with treatment and diets combined for 9 weeks. Weekly body weights, fasting blood glucose (FBG), oral glucose tolerance test (OGTT); lipid profiles, liver weights, hepatic triglycerides and adiposity were assessed at week 24.

### Results

At week 15, body weights increased between the diet group in phase 1(standard 146 ± 1.64 vs. 273.1 ± 1.56 g), (NPHC, 143.5 ± 2.40 vs. 390.2 ± 4.94 g) and (LPHC, 145.5 ± 2.28 g vs. 398.3 ± 4.89 g) (p< 0.0001). A similar increase was noted in the FBG and OGTT (p< 0.0001). In phase 2, there was an increase in FBG, OGTT, body weights, lipid profile, liver weights, hepatic triglycerides, adiposity and insulin levels in group 2 and positive control in both NPHC and LPHC diet groups (p<0.0001). Growth hormone levels were decreased in Tesamorelin-free group 2 and positive control in both NPHC and LPHC (p< 0.0001).

**Funding:** The author(s) received no specific funding for this work.

**Competing interests:** The authors have declared that no competing interests exist.

## Conclusions

The obesogenic activities of the LPHC diet exceeded that of the NPHC diet and interacted with both integrase-containing and classical cART drug regimens to reproduce cART associated metabolic dysregulation. The effects were however reversed by co-administration with tesamorelin, a synthetic growth hormone releasing hormone analogue.

## Introduction

Although the advent of combined Antiretroviral Therapy (cART), has resulted in increased lifespans and quality of life, it is often associated with the development of metabolic dysregulation e.g., dyslipidemia, insulin resistance, abnormalities in glycemic control, and lipodystrophy [1]. The cART-associated metabolic dysregulation appears to be a universal characteristic associated with antiretroviral drugs with even the newer IICR being associated with these metabolic derangements [2].

The rapid rates of urbanization in sub-Saharan Africa amidst poorly performing economies has resulted in a large proportion of the urban population having limited access to social amenities and food adequate in both quality and quantity [3]. Indeed, urban diets in the low-income urban informal settings are often high calorie (high fat/high sugar) and low protein diets (high calorie protein malnutrition).

Since Sub-Saharan Africa has a high prevalence of human immunodeficiency virus and acquired immune deficiency syndrome (HIV/AIDS) a significant proportion of patients on cART would be reasonably expected to be suffering from this high calorie protein malnutrition [4]. This study investigated the relationship between diet, cART regimens and the resulting metabolic derangements in light of recent studies that have reported increased mortality after cART initiation among patients on high calorie low protein diets than in the general population in sub-Saharan Africa [5].

## Material and methods

### Diet preparation

The various diets i.e., standard rat chow (4.8% fat, 17.1% protein, 34.6% complex carbohydrates and 5.3% sucrose), normal protein high calorie/high fat (36% fat, 17.1% protein, 42% complex carbohydrates and 20% sucrose) and low protein high calorie/high fat (36% fat, 6% protein, 42% complex carbohydrates and 20% sucrose were specially formulated and manufactured by Unga group limited, Nakuru, Kenya.

### Experimental animals' selection, grouping and treatment

One hundred and twenty (120) freshly weaned Sprague-Dawley rats (6–8 weeks old) weighing approximately 150 grams, were obtained from the Kabete veterinary laboratories, Nairobi.

The animals were grouped-housed in the animal house situated within the department of medical physiology, adhering to specified ambient conditions: a room temperature ranging from $23 \pm 2°C$, relative humidity maintained at 30–50%, and a 12-hour light/day cycle. Prior to the initiation of the study, a seven-day period was dedicated to habituating the animals to both the experimenter and the environmental conditions. The study was performed in two stages. The first phase involved the investigation of the relative obesogenic nature of the three different diets (standard chow, normal protein high calorie diet and low protein high calorie

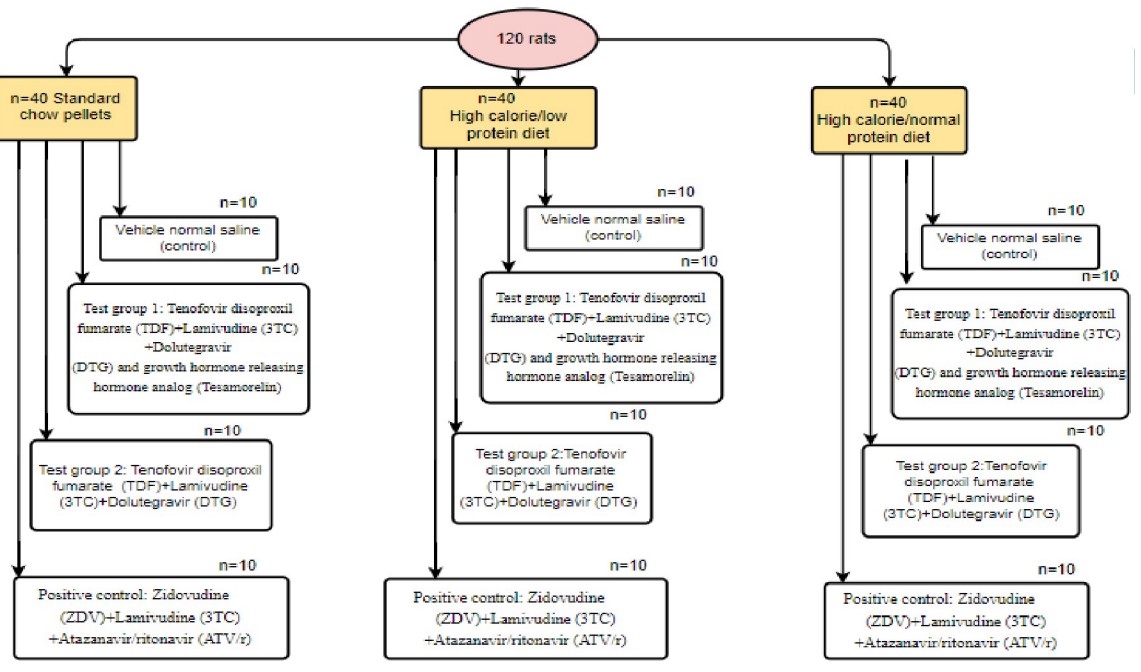

**Fig 1. A paradigm illustration of the experimental study groups.**

diet) while the second phase involved evaluating the interactions of the various treatments with the aforementioned diets.

**Phase one.** The experimental animals were randomized into the (n = 40/group) standard rat chow, normal protein/high calorie and low protein/high calorie diet groups. The respective diets and water were supplied *ad libitum* to all the groups throughout the duration of the study. This phase had a duration of fifteen weeks.

**Phase two.** The experimental animals in each diet group were further randomized into four experimental subgroups on week 16, into (n = 10/group): control (vehicle normal saline), test group 1(Tenofovir disoproxil fumarate (TDF) + Lamivudine (3TC) + Dolutegravir (DTG) + Tesamorelin, test group 2 (Tenofovir disoproxil fumarate (TDF) + Lamivudine (3TC) + Dolutegravir (DTG), and Positive control (Zidovudine (AZT) +Lamivudine (3TC) +- Atazanavir/ritonavir (ATV/r)). A paradigm illustration of the experimental study groups are shown in Fig 1.

The respective treatments were administered daily between 1500hrs and 1700hrs via oral gavage for nine (9) weeks. The dosage calculations were made using the rat to human body weight/ surface area normalization formula for drug dose calculations [6].

$$HED\,(mg\,/kg) = Animal\,dose\,(mg/kg) \times (Animal\,K_m/Human\,K_m)$$

HED is Human Equivalent Dose, Km is Correction factor

$$Rat\,K_m = 6.2, Human\,K_m = 37$$

The animals were weighed weekly using a standard laboratory weighing scale (Ohaus®SJX6201N/E scout portable balance).

## Serum biochemistry and other assays

**Fasting blood glucose and oral glucose tolerance test determination.** Weekly fasting blood glucose (FBG) levels were assessed using a glucometer (On Call® EZ II) throughout the study. Blood samples were obtained via lateral tail vein blood sampling after a six-hour fast, following the application of Topical Lidocaine ten (10) minutes prior to mitigate pain and stress associated with the test, using the Lee and Goosens protocol [7].

Oral glucose tolerance tests were conducted in the 15th and 24th weeks of the experimental period, following Bartoli's protocol [8]. Briefly, rats underwent a six-hour fast before baseline blood glucose levels were determined using the previously described procedure. Subsequently, each rat received a loading dose of glucose (2 g/kg) via oral gavage. Blood glucose levels were then measured at 30, 60, 90, and 120 minutes post-administration of the glucose load. The blood glucose levels obtained were used to calculate the area under the curves (AUCs).

**Fasting plasma insulin and growth hormone levels.** The fasting plasma insulin and growth hormone levels were determined using the enzyme-linked immunosorbent assay (ELISA) method using a rat insulin and growth hormone kit (Bioassay Technology laboratory, Shanghai, China). The fasting insulin levels was used for determination of Homeostatic Model Assessment (HOMA) score for insulin resistance and β-cell function (HOMA-IR and HOMA-β) which were calculated using the following equation [9].

$$HOMA - IR = \frac{Insulin\ (U/I) \times Blood\ glucose(mmol/I)}{22.5}$$

$$HOMA - \beta = \frac{20 \times Insulin\ (U/I)}{Blood\ glucose\ (mmol/I)} - 3.5$$

**Lipid profile and adipose tissue depot weight determination.** The rats underwent euthanasia following an overnight fasting period, achieved through intraperitoneal administration of 6% Phenobarbital on week 24. Subsequently, blood samples were obtained via cardiac puncture, left to clot, and then subjected to centrifugation at 1500 revolutions per minute for ten (10) minutes. The resulting serum was transferred into vacutainers and transported to the Department of Clinical Chemistry at the University of Nairobi. In this department, the levels of serum triglycerides, total cholesterol, low-density lipoprotein, and high-density lipoprotein were determined.

The various visceral adipose tissues depots (retroperitoneal adipose tissue, mesenteric adipose tissue and pericardial adipose tissue) were carefully extracted and weighed after euthanasia of the experimental animal.

**Determination of liver weights and hepatic triglycerides.** Following the euthanization of the experimental animals, as detailed earlier, a midline incision was performed on the ventral surface of each rat's body to expose the abdominal cavity, and the liver was subsequently excised. The respective liver weights were determined and recorded for the assessment of hepatic triglycerides. The determination of hepatic triglycerides followed the procedure outlined by Bulter and Mailing [10]. In brief, 2 grams of the respective livers were homogenized in eight milliliters of phosphate buffer. A resulting 1-milliliter portion of the homogenate was added to four grams of activated charcoal, pre-moistened with two milliliters of chloroform. After topping up the mixture with eighteen milliliters of chloroform, it was gently shaken for ten minutes, followed by filtration.

The resulting filtrate was divided into three test tubes, and an additional 1-milliliter portion of standard oil solution (1%) was pipetted into three separate test tubes. All test tubes were

placed in a water bath at 80˚C to evaporate excess chloroform. To the first and second tubes, 0.5 milliliters of alcoholic potassium hydroxide were added, and the third tube containing the filtrate and the test tube with the standard corn oil solution received 0.5 milliliters of 95% alcohol. The test tubes were maintained in water at 60˚C for twenty minutes, followed by the addition of 0.5 milliliters of 0.2N sulphuric acid to each tube. The resulting mixtures were heated in a water bath (100˚C) for an additional twenty minutes, cooled, and then subjected to the addition of 0.1 milliliter sodium metaperiodate and 0.1 milliliter sodium arsenide. Five milliliters of chromotropic acid were added to each test tube after ten minutes, and the tubes were placed in a water bath (100˚C) for half an hour. The optical densities at 540 nm were determined using a spectrophotometer. The obtained optical densities were utilized to calculate hepatic triglyceride content through the following formula:

Let

$$R = \frac{\text{Optical density (O.D)saponified unknown} - \text{O.D unsaponified unknown}}{\text{O.D saponified corn oil standard} - \text{O.D unsaponified corn oil standard}}$$

And A = volume of aliquot of chloroform extract in ml (1 ml was used in the present study).

Then triglyceride contents in milligram per gram of tissue

$$\frac{200}{A} \times R \times 0.05 = 10\frac{R}{A}$$

## Ethical considerations

The experimental protocol was approved by Biosafety, Animal Use and Ethics Committee, Faculty of Veterinary Medicine, University of Nairobi (permit number FVM BAUEC/2022/354). All surgery was performed under sodium pentobarbital anesthesia, and all efforts were made to minimize suffering.

## Statistical analysis

The experimental data were presented as mean ± standard error of the mean (S.E.M.), and statistical analysis was conducted through one-way ANOVA. In cases of significance (defined as $p \leq 0.05$), Tukey's test was applied. The analysis was carried out using GraphPad Prism® version 8.0.1(244).

## Results

### Phase one

**Body weight during the diet induction phase.** There were no significant differences in the body weight between the three experimental groups at the beginning of the study [146 ± 1.64 grams (standard diet) vs.143.5 ± 2.40 grams (normal protein high calorie diet) vs.145.5 ± 2.28 grams (low protein high calorie diet): p = 0.5538] up to the end of week 4: [179 ± 0.75 grams (standard diet) vs.181 ± 1.01 grams (normal protein high calorie diet) vs.182.6 ± 1.38 grams (low protein high calorie diet): p = 0.0688].

There were significant differences in the body weight between the three experimental groups at the end of week 5: [183 ± 1.24 grams (standard diet) vs.196 ± 1.30 grams (normal protein high calorie diet) vs. 198.6 ± 0.61 grams (low protein high calorie diet): p< 0.0001]. Post-hoc statistical analysis using Tukey's multiple comparisons test revealed significant

differences between standard diet and normal protein high calorie diet (p< 0.0001) and standard diet and low protein high calorie diet (p< 0.0001).

There were significant differences in the body weight between the three experimental groups at the end of week 6: [188.7 ± 1.33 grams (standard diet) vs. 216 ± 4.27 grams (normal protein high calorie diet) vs. 219.6 ± 1.53 grams (low protein high calorie diet): p< 0.0001]. Post-hoc statistical analysis using Tukey's multiple comparisons test revealed significant differences between standard diet and normal protein high calorie diet (p< 0.0001) and standard diet and low protein high calorie diet (p< 0.0001).

There were significant differences in the body weight between the three experimental groups at the end of week 7: [197.7 ± 1.43 grams (standard diet) vs. 230 ± 2.70 grams (normal protein high calorie diet) vs. 233.3 ± 1.52 grams (low protein high calorie diet): p< 0.0001]. Post-hoc statistical analysis using Tukey's multiple comparisons test revealed significant differences between standard diet and normal protein high calorie diet (p< 0.0001) and standard diet and low protein high calorie diet (p< 0.0001).

There were significant differences in the body weight between the three experimental groups at the end of week 8: [208.7 ± 1.27 grams (standard diet) vs. 255 ± 2.69 grams (normal protein high calorie diet) vs. 258 ± 1.46 grams (low protein high calorie diet): p< 0.0001]. Post-hoc statistical analysis using Tukey's multiple comparisons test revealed significant differences between standard diet and normal protein high calorie diet (p< 0.0001) and standard diet and low protein high calorie diet (p< 0.0001).

There were significant differences in the body weight between the three experimental groups at the end of week 9: [218 ± 2.44 grams (standard diet) vs. 277 ± 7.06 grams (normal protein high calorie diet) vs. 283 ± 4.09 grams (low protein high calorie diet): p< 0.0001]. Post-hoc statistical analysis using Tukey's multiple comparisons test revealed significant differences between standard diet and normal protein high calorie diet (p< 0.0001) and standard diet and low protein high calorie diet (p< 0.0001).

There were significant differences in the body weight between the three experimental groups at the end of week 10: [221 ± 1.40 grams (standard diet) vs. 286.8 ± 3.20 grams (normal protein high calorie diet) vs. 292 ± 3.77 grams (low protein high calorie diet): p< 0.0001]. Post-hoc statistical analysis using Tukey's multiple comparisons test revealed significant differences between standard diet and normal protein high calorie diet (p< 0.0001) and standard diet and low protein high calorie diet (p< 0.0001).

There were significant differences in the body weight between the three experimental groups at the end of week 11: [228.6 ± 1.22 grams (standard diet) vs. 302 ± 4.88 grams (normal protein high calorie diet) vs. 307 ± 4.89 grams (low protein high calorie diet): p< 0.0001]. Post-hoc statistical analysis using Tukey's multiple comparisons test revealed significant differences between standard diet and normal protein high calorie diet (p< 0.0001) and standard diet and low protein high calorie diet (p< 0.0001).

There were significant differences in the body weight between the three experimental groups at the end of week 12: [237 ± 1.31 grams (standard diet) vs. 322 ± 5.47 grams (normal protein high calorie diet) vs. 329 ± 2.93 grams (low protein high calorie diet): p< 0.0001]. Post-hoc statistical analysis using Tukey's multiple comparisons test revealed significant differences between standard diet and normal protein high calorie diet (p< 0.0001) and standard diet and low protein high calorie diet (p< 0.0001).

There were significant differences in the body weight between the three experimental groups at the end of week 13: [246.6 ± 1.44 grams (standard diet) vs. 341.2 ± 5.15 grams (normal protein high calorie diet) vs. 352.2 ± 6.22 grams (low protein high calorie diet): p< 0.0001]. Post-hoc statistical analysis using Tukey's multiple comparisons test revealed

significant differences between standard diet and normal protein high calorie diet (p< 0.0001) and standard diet and low protein high calorie diet (p< 0.0001).

There were significant differences in the body weight between the three experimental groups at the end of week 14: [256.4 ± 1.49 grams (standard diet) vs. 360.2 ± 5.60 grams (normal protein high calorie diet) vs. 372.3 ± 3.48 grams (low protein high calorie diet): p< 0.0001]. Post-hoc statistical analysis using Tukey's multiple comparisons test revealed significant differences between standard diet and normal protein high calorie diet (p< 0.0001) and standard diet and low protein high calorie diet (p< 0.0001).

There were significant differences in the body weight between the three experimental groups at the end of week 15: [273.1 ± 1.56 grams (standard diet) vs. 390.2 ± 4.94 grams (normal protein high calorie diet) vs. 398.3 ± 4.89 grams (low protein high calorie diet): p< 0.0001]. Post-hoc statistical analysis using Tukey's multiple comparisons test revealed significant differences between standard diet and normal protein high calorie diet (p< 0.0001) and standard diet and low protein high calorie diet (p< 0.0001).

The graphical presentation of the mean body weights at weekly interval during the diet induction phase is shown in Fig 2 (line graph) and Table 1.

**Fasting blood glucose.** There were no significant differences in the fasting blood glucose between the three experimental groups at the beginning of the experiment [3.86 ± 0.04 mmol/L (standard diet) vs. 3.85 ± 0.03 mmol/L (normal protein high calorie diet) vs. 3.85 ± 0.18 mmol/L (low protein high calorie diet): p = 0.9996] up to the end of week 4: [4.00 ± 0.02 mmol/L (standard diet) vs. 4.03 ± 0.02 mmol/L (normal protein high calorie diet) vs. 4.06 ± 0.02 mmol/L (low protein high calorie diet): p = 0.3748].

There were significant differences in the fasting blood glucose between the three experimental groups at the at the end of week 5: [4.00 ± 0.04 mmol/L (standard diet) vs. 4.08 ± 0.02 mmol/L (normal protein high calorie diet) vs. 4.09 ± 0.01 mmol/L (low protein high calorie diet): p< 0.0001]. Post-hoc statistical analysis using Tukey's multiple comparisons test revealed significant differences between standard diet and normal protein high calorie diet (p = 0.0013) and standard diet and low protein high calorie diet (p = 0.0004).

There were significant differences in the fasting blood glucose between the three experimental groups at the at the end of week 6: [4.05 ± 0.02 mmol/L (standard diet) vs. 4.14 ± 0.02 mmol/L (normal protein high calorie diet) vs. 4.20 ± 0.01 mmol/L (low protein high calorie diet): p< 0.0001]. Post-hoc statistical analysis using Tukey's multiple comparisons test revealed significant differences between standard diet and normal protein high calorie diet (p = 0.0003) and standard diet and low protein high calorie diet (p< 0.0001).

There were significant differences in the fasting blood glucose between the three experimental groups at the at the end of week 7: [4.06 ± 0.02 mmol/L (standard diet) vs. 4.18 ± 0.02 mmol/L (normal protein high calorie diet) vs. 4.23 ± 0.02 mmol/L (low protein high calorie diet): p< 0.0001]. Post-hoc statistical analysis using Tukey's multiple comparisons test revealed significant differences between standard diet and normal protein high calorie diet (p< 0.0001) and standard diet and low protein high calorie diet (p< 0.0001).

There were significant differences in the fasting blood glucose between the three experimental groups at the at the end of week 8: [4.02 ± 0.02 mmol/L (standard diet) vs. 4.23 ± 0.02 mmol/L (normal protein high calorie diet) vs. 4.32 ± 0.02 mmol/L (low protein high calorie diet): p< 0.0001]. Post-hoc statistical analysis using Tukey's multiple comparisons test revealed significant differences between standard diet and normal protein high calorie diet (p< 0.0001) and standard diet and low protein high calorie diet (p< 0.0001).

There were significant differences in the fasting blood glucose between the three experimental groups at the at the end of week 9: [4.08 ± 0.02 mmol/L (standard diet) vs. 4.37 ± 0.02 mmol/L (normal protein high calorie diet) vs. 4.43 ± 0.02 mmol/L (low protein high calorie

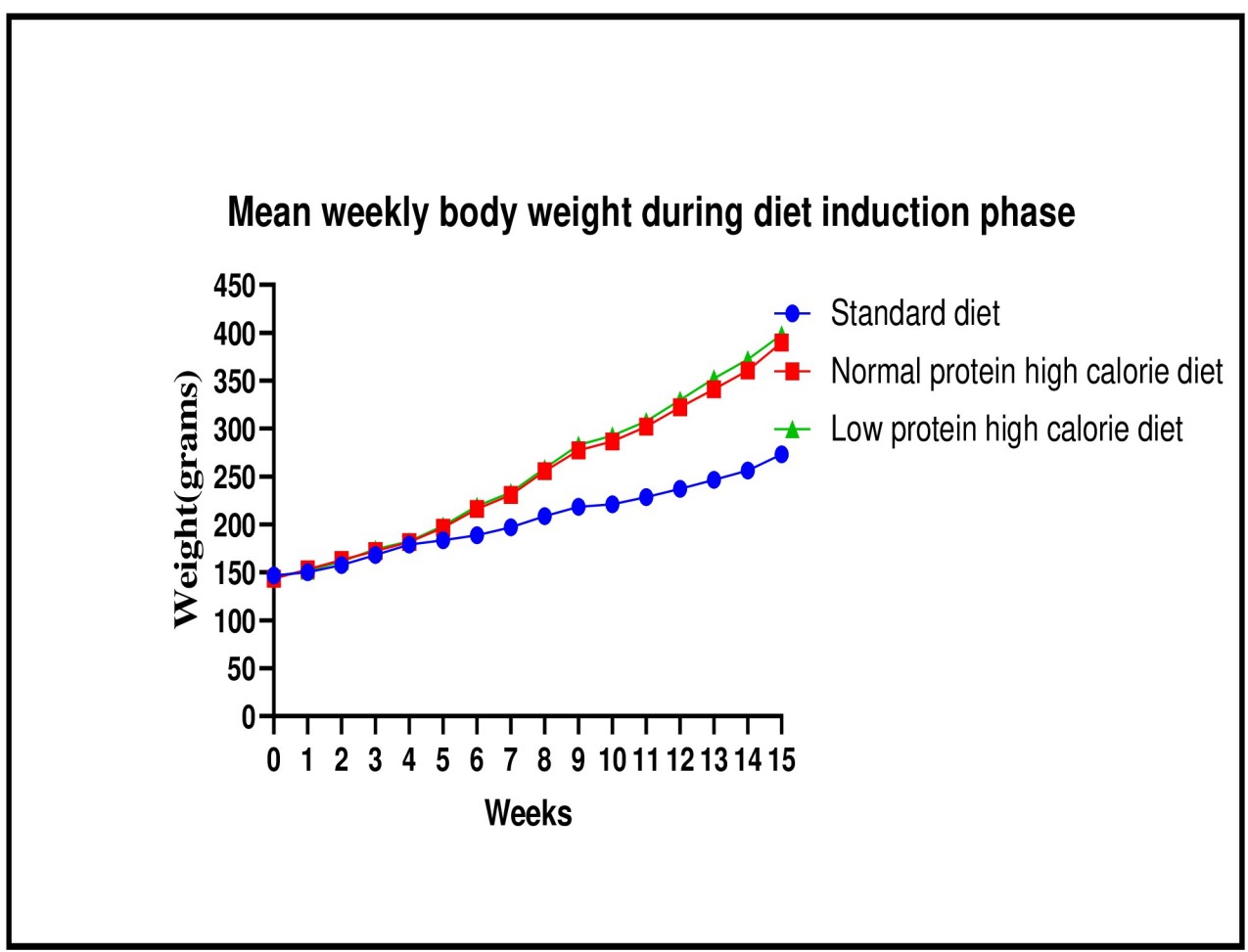

**Fig 2. Line graph showing the mean body weights (g) at weekly interval during the diet induction phase.** Results expressed as mean± SEM.

diet): p< 0.0001]. Post-hoc statistical analysis using Tukey's multiple comparisons test revealed significant differences between standard diet and normal protein high calorie diet (p< 0.0001) and standard diet and low protein high calorie diet (p< 0.0001).

There were significant differences in the fasting blood glucose between the three experimental groups at the at the end of week 10: [4.12 ± 0.02 mmol/L (standard diet) vs. 4.45 ± 0.02 mmol/L (normal protein high calorie diet) vs. 4.50 ± 0.02 mmol/L (low protein high calorie diet): p< 0.0001]. Post-hoc statistical analysis using Tukey's multiple comparisons test revealed significant differences between standard diet and normal protein high calorie diet (p< 0.0001) and standard diet and low protein high calorie diet (p< 0.0001).

There were significant differences in the fasting blood glucose between the three experimental groups at the at the end of week 11: [4.10 ± 0.01 mmol/L (standard diet) vs. 4.57 ± 0.03 mmol/L (normal protein high calorie diet) vs. 4.86 ± 0.03 mmol/L (low protein high calorie diet): p< 0.0001]. Post-hoc statistical analysis using Tukey's multiple comparisons test revealed significant differences between standard diet and normal protein high calorie diet (p< 0.0001) and standard diet and low protein high calorie diet (p< 0.0001).

There were significant differences in the fasting blood glucose between the three experimental groups at the at the end of week 12: [4.11 ± 0.02 mmol/L (standard diet) vs. 5.10 ± 0.03

**Table 1. Mean body weights (g) during the diet induction phase.**

| Groups | Standard diet | Normal protein high calorie diet | Low protein high calorie diet | P = Value |
|---|---|---|---|---|
| **Body weights (g) Phase 1 (Diet induction phase)** | | | | |
| Number of rats | n = 40 | n = 40 | n = 40 | |
| Week 0 (Baseline) | 146.8±1.642 | 143.5±2.400 | 145.5±2.283 | 0.5538 |
| Week 1 | 150.1±1.402 | 153.0±1.132 | 151.5±0.9886 | 0.2254 |
| Week 2 | 158.0±1.425 | 163.1±1.693 | 161.8±1.499 | 0.0613 |
| Week 3 | 168.4±0.938 | 172.3±2.816 | 174.1±1.295 | 0.0910 |
| Week 4 | 179.1±0.753 | 181.7±1.005 | 182.6±1.377 | 0.0688 |
| Week 5 | 183.4±1.242 | 196.8±1.295 | 198.6±0.609 | <0.0001**** |
| Week 6 | 188.7±1.325 | 216.0±4.265 | 219.3±1.526 | <0.0001**** |
| Week 7 | 197.0±1.431 | 230.7±2.703 | 233.3±1.523 | <0.0001**** |
| Week 8 | 208.8±1.274 | 255.4±2.689 | 258.5±1.460 | <0.0001**** |
| Week 9 | 218.3±2.435 | 277.4±7.056 | 283.1±4.089 | <0.0001**** |
| Week 10 | 221.2±1.395 | 286.8±3.203 | 292.8±3.764 | <0.0001**** |
| Week 11 | 228.6±1.219 | 302.0±4.882 | 307.8±4.884 | <0.0001**** |
| Week 12 | 237.0±1.312 | 322.3±5.465 | 329.8±2.931 | <0.0001**** |
| Week 13 | 246.6±1.436 | 341.2±5.146 | 352.2±6.219 | <0.0001**** |
| Week 14 | 256.4±1.488 | 360.7±5.601 | 372.3±3.476 | <0.0001**** |
| Week 15 | 273.1±1.558 | 390.0±4.936 | 398.1±4.888 | <0.0001**** |

mmol/L (normal protein high calorie diet) vs. 5.34 ± 0.04 mmol/L (low protein high calorie diet): $p < 0.0001$]. Post-hoc statistical analysis using Tukey's multiple comparisons test revealed significant differences between standard diet and normal protein high calorie diet ($p < 0.0001$) and standard diet and low protein high calorie diet ($p < 0.0001$).

There were significant differences in the fasting blood glucose between the three experimental groups at the at the end of week 13: [4.14 ± 0.02 mmol/L (standard diet) vs. 5.65 ± 0.04 mmol/L (normal protein high calorie diet) vs. 5.96 ± 0.04 mmol/L (low protein high calorie diet): $p < 0.0001$]. Post-hoc statistical analysis using Tukey's multiple comparisons test revealed significant differences between standard diet and normal protein high calorie diet ($p < 0.0001$) and standard diet and low protein high calorie diet ($p < 0.0001$).

There were significant differences in the fasting blood glucose between the three experimental groups at the at the end of week 14: [4.05 ± 0.02 mmol/L (standard diet) vs. 5.72 ± 0.05 mmol/L (normal protein high calorie diet) vs. 6.26 ± 0.30 mmol/L (low protein high calorie diet): $p < 0.0001$]. Post-hoc statistical analysis using Tukey's multiple comparisons test revealed significant differences between standard diet and normal protein high calorie diet ($p < 0.0001$) and standard diet and low protein high calorie diet ($p < 0.0001$).

There were significant differences in the fasting blood glucose between the three experimental groups at the at the end of week 15: [4.05 ± 0.02 mmol/L (standard diet) vs. 5.83 ± 0.05 mmol/L (normal protein high calorie diet) vs. 6.56 ± 0.30 mmol/L (low protein high calorie diet): $p < 0.0001$]. Post-hoc statistical analysis using Tukey's multiple comparisons test revealed significant differences between standard diet and normal protein high calorie diet ($p < 0.0001$) and standard diet and low protein high calorie diet ($p < 0.0001$). The graphical representations of the experimental data are shown in Fig 3.

**Oral glucose tolerance test at week 15 (Diet induction phase).** There were significant differences in the AUC values between the three experimental groups on week 15: [530 ± 2.31 mmol/L.min (standard diet) vs. 768.4 ± 4.03 mmol/L.min (normal protein high calorie diet) vs. 927.9 ± 2.80 mmol/L.min (low protein high calorie diet): $p < 0.0001$]. Post-hoc statistical

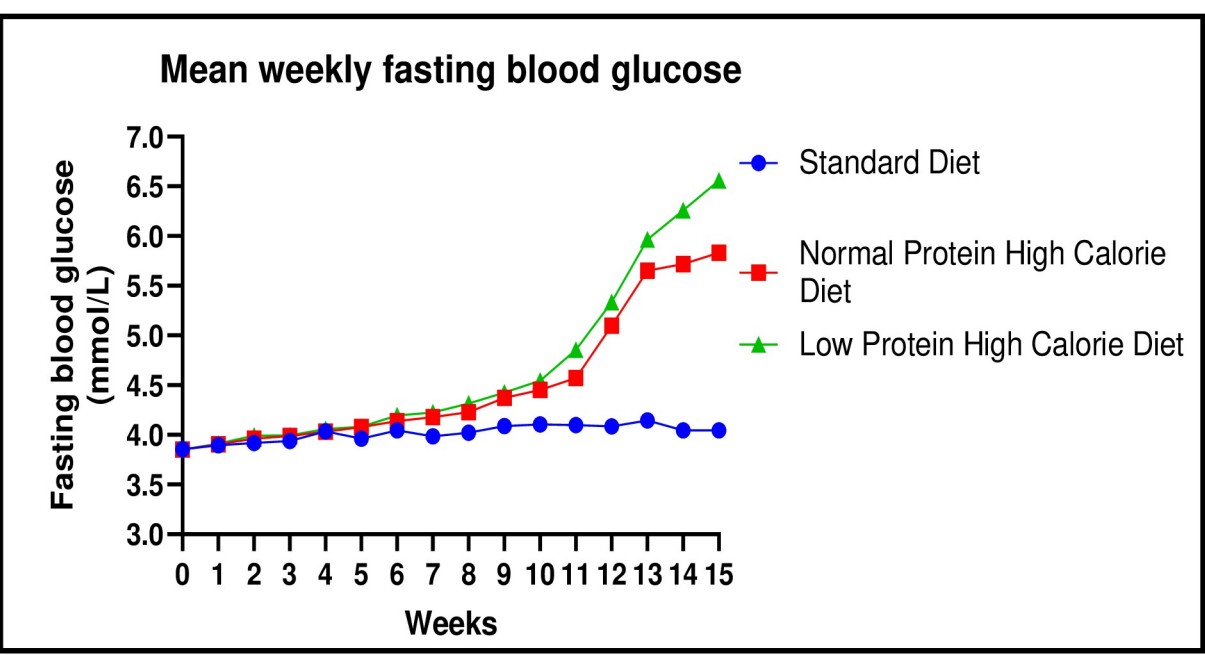

**Fig 3. Line graph showing mean fasting blood glucose levels (mmol/l) at weekly intervals during the diet induction phase.**

analysis using Tukey's multiple comparisons test revealed significant differences between standard diet and normal protein high calorie diet (p< 0.0001), standard diet and low protein high calorie diet (p< 0.0001) and, normal protein high calorie diet and low protein high calorie diet (p< 0.0001).

The graphical presentation of the mean blood glucose response and mean area under the curve during the diet induction phase is shown in Fig 4.

### Treatment phase

**Body weights during treatment phase.** *Standard diet.* There were no significant differences in the body weight between the four experimental groups at the end of week 16: [277.2 ± 5.31 grams (normal saline) vs. 284 ± 3.38 grams (Test group 1) vs. 286.6 ± 2.43 grams (Test group 2) vs. 288 ± 3.39 grams (positive control): p = 0.2011].

There were no significant differences in the body weight between the four experimental groups at the end of week 17: [292.4 ± 2.73 grams (normal saline) vs. 293.8 ± 3.68 grams (Test group 1) vs. 301.2 ± 2.41 grams (Test group 2) vs. 302.1 ± 3.10 grams (positive control): p = 0.0612].

There were no significant differences in the body weight between the four experimental groups at the end of week 18: [313.1 ± 1.79 grams (normal saline) vs. 315.4 ± 0.99 grams (Test group 1) vs. 318.5 ± 1.88 grams (Test group 2) vs. 320.4 ± 3.24 grams (positive control): p = 0.0900].

There were no significant differences in the body weight between the four experimental groups at the end of week 19: [319.9 ± 0.80 grams (normal saline) vs. 318.0 ± 1.19 grams (Test group 1) vs. 320.5 ± 1.65 grams (Test group 2) vs. 322.1 ± 1.09 grams (positive control): p = 0.1520].

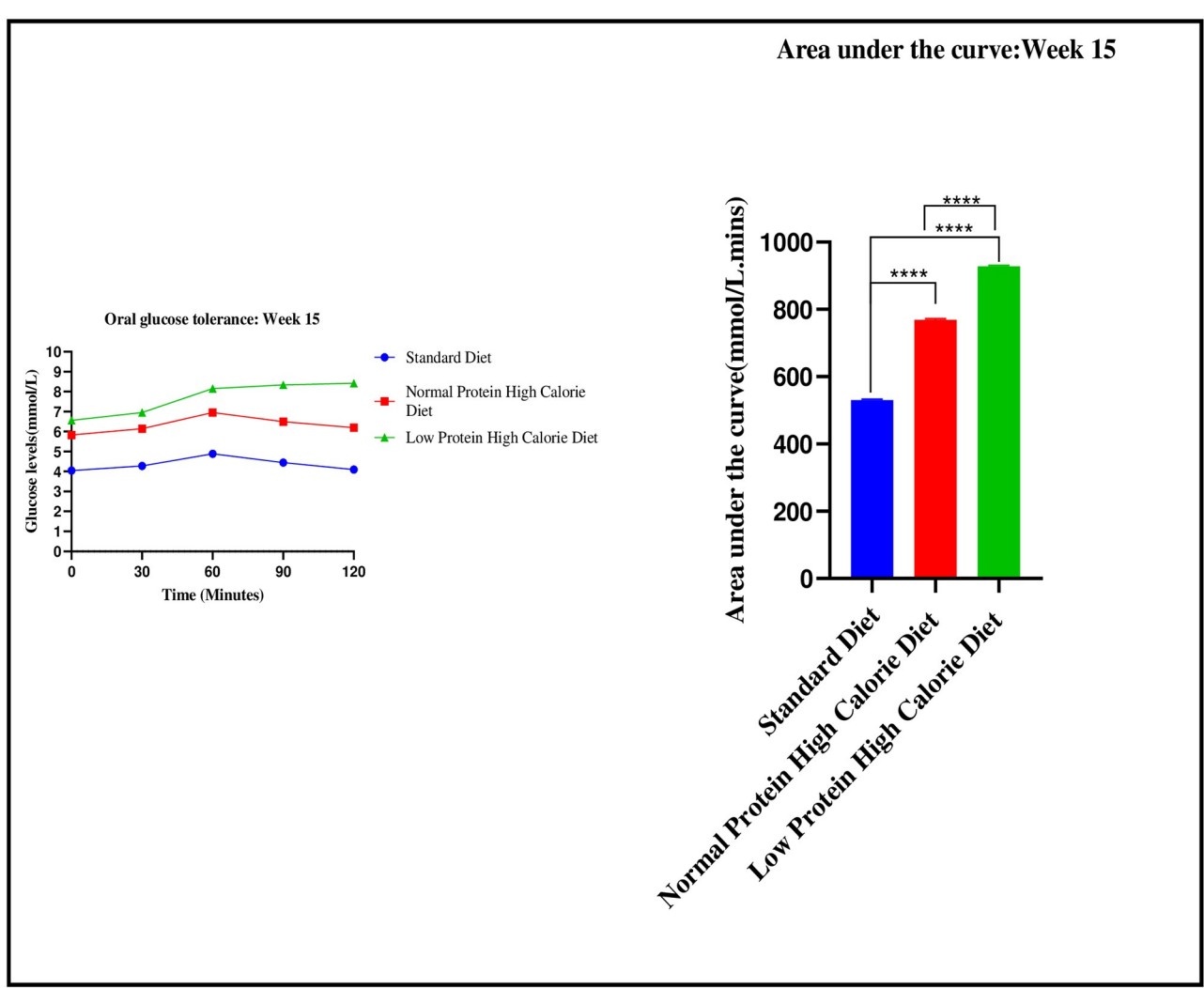

**Fig 4. Mean blood glucose response (mmol/L) to an oral glucose bolus 2 g/kg over a 2-hour period and mean area under the curve (mmol/L.min) during the oral glucose tolerance test.** Results are expressed as mean ± SEM. ****- p < 0.0001.

There were no significant differences in the body weight between the four experimental groups at the end of week 20: [329.4 ± 1.05 grams (normal saline) vs. 327.1 ± 0.87 grams (Test group 1) vs. 330.5 ± 0.54 grams (Test group 2) vs. 330.8 ± 1.60 grams (positive control): p = 0.0901].

There were no significant differences in the body weight between the four experimental groups at the end of week 21: [339.3 ± 1.11 grams (normal saline) vs. 341.0 ± 1.02 grams (Test group 1) vs. 343.6 ± 0.94 grams (Test group 2) vs. 343.6 ± 0.94 grams (positive control): p = 0.0790].

There were no significant differences in the body weight between the four experimental groups at the end of week 22: [345.7 ± 1.10 grams (normal saline) vs. 346.3 ± 1.37 grams (Test group 1) vs. 362.9 ± 1.44 grams (Test group 2) vs. 349 ± 1.82 grams (positive control): p = 0.0521].

There were no significant differences in the body weight between the four experimental groups at the end of week 23: [360 ± 0.79 grams (normal saline) vs. 360.6 ± 1.44 grams (Test

group 1) vs. 364.3 ± 1.37 grams (Test group 2) vs. 362.9 ± 1.12 grams (positive control): p = 0.0531].

There were no significant differences in the body weight between the four experimental groups at the end of week 24: [374.8 ± 1.36 grams (normal saline) vs. 375.2 ± 1.66 grams (Test group 1) vs. 378.6 ± 1.06 grams (Test group 2) vs. 377.1 ± 0.87 grams (positive control): p = 0.1534].

*Normal protein high calorie diet*. There were no significant differences in the body weight between the four experimental groups at the end of week 16: [418.3 ± 4.83 grams (normal saline) vs. 414.1 ± 6.09 grams (Test group 1) vs. 412 ± 7.72 grams (Test group 2) vs. 415 ± 5.05 grams (positive control): p = 0.1917].

There were no significant differences in the body weight between the four experimental groups at the end of week 17: [419.9 ± 3.32 grams (normal saline) vs. 413.9 ± 9.15 grams (Test group 1) vs. 431.6 ± 7.68 grams (Test group 2) vs. 435 ± 5.60 grams (positive control): p = 0.1092].

There were significant differences in the body weight between the four experimental groups at the end of week 18: [424.2 ± 4.02 grams (normal saline) vs. 426.4 ± 5.47 grams (Test group 1) vs. 453 ± 7.74 grams (Test group 2) vs. 456 ± 5.82 grams (positive control): p = 0.0002]. Post-hoc statistical analysis using Tukey's multiple comparisons test revealed significant differences between normal saline and Test group 2 (p = 0.0059), normal saline and positive control (p = 0.0026), Test group 1 and Test group 2 (p = 0.0117) and, Test group 1 and positive control (p = 0.0053).

There were significant differences in the body weight between the four experimental groups at the end of week 19: [433.2 ± 4.22 grams (normal saline) vs. 431.2 ± 4.40 grams (Test group 1) vs. 470 ± 7.27 grams (Test group 2) vs. 477 ± 5.73 grams (positive control): p < 0.0001]. Post-hoc statistical analysis using Tukey's multiple comparisons test revealed significant differences between normal saline and Test group 2 (p = 0.0002), normal saline and positive control (p < 0.0001), Test group 1 and Test group 2 (p < 0.0001) and, Test group 1 and positive control (p < 0.0001).

There were significant differences in the body weight between the four experimental groups at the end of week 20: [441.5 ± 4.62 grams (normal saline) vs. 438.5 ± 4.32 grams (Test group 1) vs. 489.1 ± 6.87 grams (Test group 2) vs. 497 ± 5.73 grams (positive control): p < 0.0001]. Post-hoc statistical analysis using Tukey's multiple comparisons test revealed significant differences between normal saline and Test group 2 (p < 0.0001), normal saline and positive control (p < 0.0001), Test group 1 and Test group 2 (p < 0.0001) and, Test group 1 and positive control (p < 0.0001).

There were significant differences in the body weight between the four experimental groups at the end of week 21: [450.5 ± 4.54 grams (normal saline) vs. 448 ± 4.15 grams (Test group 1) vs. 510.8 ± 6.51 grams (Test group 2) vs. 518.7 ± 5.75 grams (positive control): p < 0.0001]. Post-hoc statistical analysis using Tukey's multiple comparisons test revealed significant differences between normal saline and Test group 2 (p < 0.0001), normal saline and positive control (p < 0.0001), Test group 1 and Test group 2 (p < 0.0001) and, Test group 1 and positive control (p < 0.0001).

There were significant differences in the body weight between the four experimental groups at the end of week 22: [460.4 ± 4.63 grams (normal saline) vs. 461.3 ± 4.14 grams (Test group 1) vs. 532.8 ± 6.83 grams (Test group 2) vs. 539 ± 5.93 grams (positive control): p < 0.0001]. Post-hoc statistical analysis using Tukey's multiple comparisons test revealed significant differences between normal saline and Test group 2 (p < 0.0001), normal saline and positive control (p < 0.0001), Test group 1 and Test group 2 (p < 0.0001) and, Test group 1 and positive control (p < 0.0001).

There were significant differences in the body weight between the four experimental groups at the end of week 23: [469.2 ± 4.85 grams (normal saline) vs. 466.3 ± 2.62 grams (Test group 1) vs. 543.4 ± 13.6 grams (Test group 2) vs. 550.3 ± 13.72 grams (positive control): p< 0.0001]. Post-hoc statistical analysis using Tukey's multiple comparisons test revealed significant differences between normal saline and Test group 2 (p< 0.0001), normal saline and positive control (p< 0.0001), Test group 1 and Test group 2 (p< 0.0001) and, Test group 1 and positive control (p< 0.0001).

There were significant differences in the body weight between the four experimental groups at the end of week 24: [478.4 ± 4.53 grams (normal saline) vs. 479.9 ± 4.93 grams (Test group 1) vs. 564.3 ± 13.69 grams (Test group 2) vs. 582.4 ± 5.37 grams (positive control): p< 0.0001]. Post-hoc statistical analysis using Tukey's multiple comparisons test revealed significant differences between normal saline and Test group 2 (p< 0.0001), normal saline and positive control (p< 0.0001), Test group 1 and Test group 2 (p< 0.0001) and, Test group 1 and positive control (p< 0.0001).

*Low protein high calorie diet*. There were no significant differences in the body weight between the four experimental groups at the end of week 16: [413.9 ± 4.42 grams (normal saline) vs. 410.6 ± 6.79 grams (Test group 1) vs. 419.4 ± 6.98 grams (Test group 2) vs. 419.6 ± 3.91 grams (positive control): p = 0.6240].

There were no significant differences in the body weight between the four experimental groups at the end of week 17: [433.2 ± 3.92 grams (normal saline) vs. 428.5 ± 5.69 grams (Test group 1) vs. 444.2 ± 7.21 grams (Test group 2) vs. 442.6 ± 3.80 grams (positive control): p = 0.1312].

There were significant differences in the body weight between the four experimental groups at the end of week 18: [442.1 ± 4.08 grams (normal saline) vs. 437.8 ± 5.96 grams (Test group 1) vs. 465.7 ± 7.54 grams (Test group 2) vs. 464.1 ± 3.58 grams (positive control): p = 0.0008]. Post-hoc statistical analysis using Tukey's multiple comparisons test revealed significant differences between normal saline and Test group 2 (p = 0.0226), normal saline and positive control (p = 0.0382), Test group 1 and Test group 2 (p = 0.0053) and, Test group 1 and positive control (p = 0.0095).

There were significant differences in the body weight between the four experimental groups at the end of week 19: [448 ± 3.18 grams (normal saline) vs. 450.4 ± 4.90 grams (Test group 1) vs. 482.2 ± 6.53 grams (Test group 2) vs. 488 ± 2.47 grams (positive control): p< 0.0001]. Post-hoc statistical analysis using Tukey's multiple comparisons test revealed significant differences between normal saline and Test group 2 (p< 0.0001), normal saline and positive control (p< 0.0001), Test group 1 and Test group 2 (p< 0.0001) and, Test group 1 and positive control (p< 0.0001).

There were significant differences in the body weight between the four experimental groups at the end of week 20: [458.6 ± 2.88 grams (normal saline) vs. 464.2 ± 4.62 grams (Test group 1) vs. 503.8 ± 7.49 grams (Test group 2) vs. 510.9 ± 2.78 grams (positive control): p< 0.0001]. Post-hoc statistical analysis using Tukey's multiple comparisons test revealed significant differences between normal saline and Test group 2 (p< 0.0001), normal saline and positive control (p< 0.0001), Test group 1 and Test group 2 (p< 0.0001) and, Test group 1 and positive control (p< 0.0001).

There were significant differences in the body weight between the four experimental groups at the end of week 21: [468.5 ± 3.00 grams (normal saline) vs. 475.8 ± 5.29 grams (Test group 1) vs. 520.2 ± 6.21 grams (Test group 2) vs. 531.5 ± 2.16 grams (positive control): p< 0.0001]. Post-hoc statistical analysis using Tukey's multiple comparisons test revealed significant differences between normal saline and Test group 2 (p< 0.0001), normal saline and positive control

(p< 0.0001), Test group 1 and Test group 2 (p< 0.0001) and, Test group 1 and positive control (p< 0.0001).

There were significant differences in the body weight between the four experimental groups at the end of week 22: [479.2 ± 3.29 grams (normal saline) vs. 484.2 ± 5.90 grams (Test group 1) vs. 541.9 ± 6.03 grams (Test group 2) vs. 554.5 ± 1.91 grams (positive control): p< 0.0001]. Post-hoc statistical analysis using Tukey's multiple comparisons test revealed significant differences between normal saline and Test group 2 (p< 0.0001), normal saline and positive control (p< 0.0001), Test group 1 and Test group 2 (p< 0.0001) and, Test group 1 and positive control (p< 0.0001).

There were significant differences in the body weight between the four experimental groups at the end of week 23: [488.8 ± 2.73 grams (normal saline) vs. 492.2 ± 5.79 grams (Test group 1) vs. 561 ± 5.68 grams (Test group 2) vs. 577 ± 1.98 grams (positive control): p< 0.0001]. Post-hoc statistical analysis using Tukey's multiple comparisons test revealed significant differences between normal saline and Test group 2 (p< 0.0001), normal saline and positive control (p< 0.0001), Test group 1 and Test group 2 (p< 0.0001) and, Test group 1 and positive control (p< 0.0001).

There were significant differences in the body weight between the four experimental groups at the end of week 24: [497.8 ± 3.10 grams (normal saline) vs. 501.7 ± 5.22 grams (Test group 1) vs. 589.9 ± 3.63 grams (Test group 2) vs. 600 ± 1.94 grams (positive control): p< 0.0001]. Post-hoc statistical analysis using Tukey's multiple comparisons test revealed significant differences between normal saline and Test group 2 (p< 0.0001), normal saline and positive control (p< 0.0001), Test group 1 and Test group 2 (p< 0.0001) and, Test group 1 and positive control (p< 0.0001).

The graphical presentation of the mean body weights at weekly interval during the treatment phase is shown in Fig 5.

**Fasting blood glucose during the treatment phase.** *Standard diet.* There were no significant differences in the fasting blood glucose between the four experimental groups at the end of week 16: [4.05 ± 0.03 mmol/L (normal saline) vs. 4.09 ± 0.05 mmol/L (Test group 1) vs. 4.12 ± 0.03 mmol/L (Test group 2) vs. 4.17 ± 0.04 mmol/L (positive control): p = 0.1381].

There were no significant differences in the fasting blood glucose between the four experimental groups at the end of week 17: [4.06 ± 0.03 mmol/L (normal saline) vs. 4.09 ± 0.05 mmol/L (Test group 1) vs. 4.16 ± 0.03 mmol/L (Test group 2) vs. 4.17 ± 0.03 mmol/L (positive control): p = 0.0581].

There were no significant differences in the fasting blood glucose between the four experimental groups at the end of week 18: [4.08 ± 0.04 mmol/L (normal saline) vs. 4.06 ± 0.03 mmol/L (Test group 1) vs. 4.12 ± 0.03 mmol/L (Test group 2) vs. 4.18 ± 0.03 mmol/L (positive control): p = 0.0620].

There were no significant differences in the fasting blood glucose between the four experimental groups at the end of week 19: [4.08 ± 0.03 mmol/L (normal saline) vs. 4.10 ± 0.03 mmol/L (Test group 1) vs. 4.16 ± 0.05 mmol/L (Test group 2) vs. 4.20 ± 0.03 mmol/L (positive control): p = 0.0964].

There were no significant differences in the fasting blood glucose between the four experimental groups at the end of week 20: [4.13 ± 0.05 mmol/L (normal saline) vs. 4.13 ± 0.03 mmol/L (Test group 1) vs. 4.20 ± 0.04 mmol/L (Test group 2) vs. 4.24 ± 0.04 mmol/L (positive control): p = 0.1605].

There were no significant differences in the fasting blood glucose between the four experimental groups at the end of week 21: [4.16 ± 0.04 mmol/L (normal saline) vs. 4.15 ± 0.04 mmol/L (Test group 1) vs. 4.21 ± 0.04 mmol/L (Test group 2) vs. 4.20 ± 0.02 mmol/L (positive control): p = 0.6454].

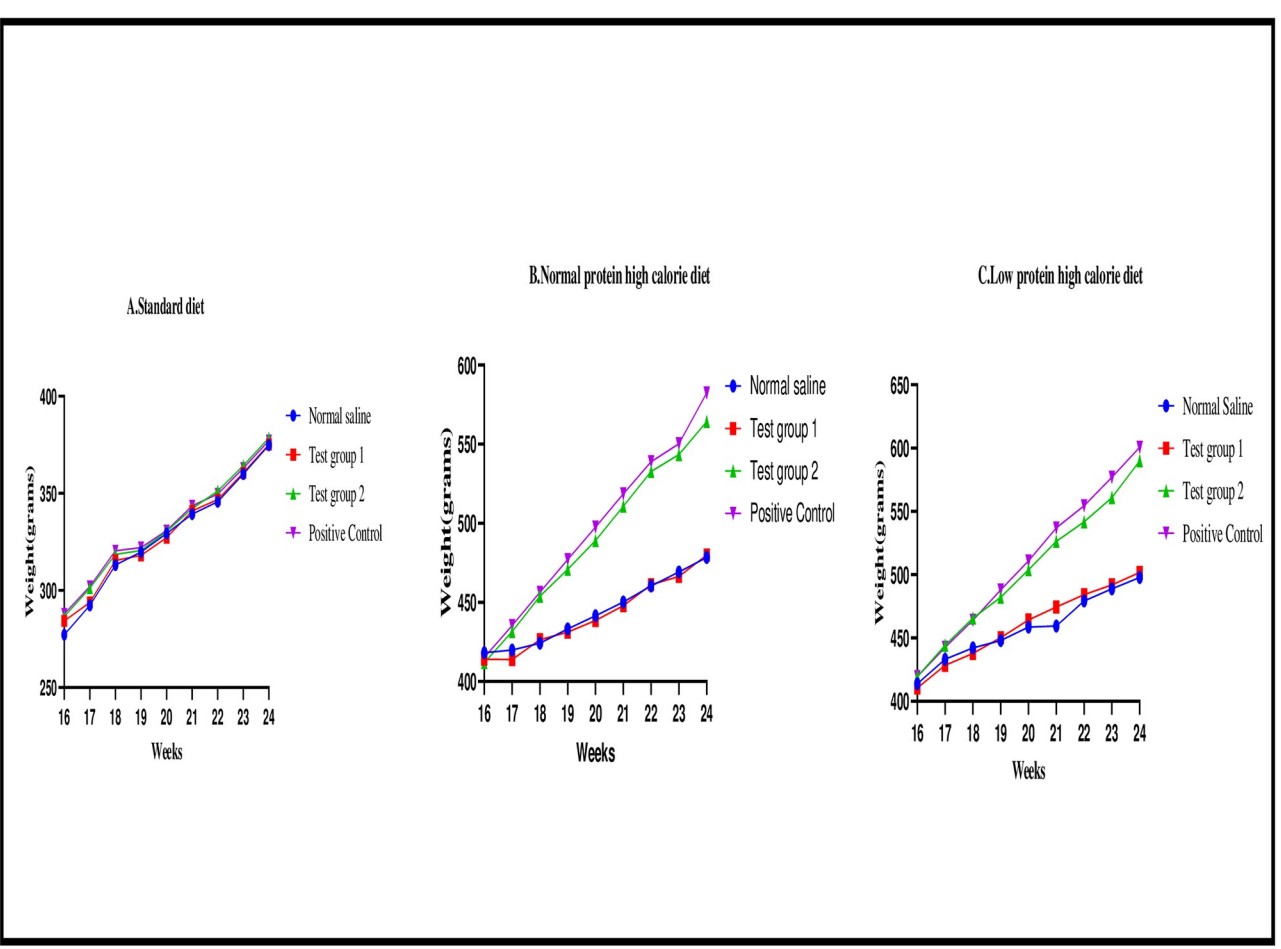

**Fig 5. Line graphs showing the mean body weights (grams) at weekly interval during the treatment phase.** Expressed as mean ± SEM. A (standard diet group), B (normal protein high calorie diet group), C (low protein high calorie diet group).

There were no significant differences in the fasting blood glucose between the four experimental groups at the end of week 22: [4.18 ± 0.03 mmol/L (normal saline) vs. 4.17 ± 0.03 mmol/L (Test group 1) vs. 4.26 ± 0.04 mmol/L (Test group 2) vs. 4.22 ± 0.03 mmol/L (positive control): p = 0.0867].

There were no significant differences in the fasting blood glucose between the four experimental groups at the end of week 23: [4.14 ± 0.03 mmol/L (normal saline) vs. 4.18 ± 0.02 mmol/L (Test group 1) vs. 4.23 ± 0.02 mmol/L (Test group 2) vs. 4.22 ± 0.04 mmol/L (positive control): p = 0.8025].

There were no significant differences in the fasting blood glucose between the four experimental groups at the end of week 24: [4.16 ± 0.03 mmol/L (normal saline) vs. 4.12 ± 0.02 mmol/L (Test group 1) vs. 4.25 ± 0.02 mmol/L (Test group 2) vs. 4.21 ± 0.03 mmol/L (positive control): p = 0.2334].

*Normal protein high calorie diet*. There were no significant differences in the fasting blood glucose between the four experimental groups at the end of week 16: [5.76 ± 0.06 mmol/L (normal saline) vs. 5.78 ± 0.07 mmol/L (Test group 1) vs. 5.84 ± 0.95 mmol/L (Test group 2) vs. 6.01 ± 0.03 mmol/L (positive control): p = 0.0637].

There were no significant differences in the fasting blood glucose between the four experimental groups at the end of week 17: [5.89 ± 0.05 mmol/L (normal saline) vs. 5.91 ± 0.05 mmol/L (Test group 1) vs. 6.04 ± 0.06 mmol/L (Test group 2) vs. 6.08 ± 0.62 mmol/L (positive control): p = 0.0535].

There were significant differences in the fasting blood glucose between the four experimental groups at the end of week 18: [6.05 ± 0.05 mmol/L (normal saline) vs. 6.06 ± 0.06 mmol/L (Test group 1) vs. 6.26 ± 0.03 mmol/L (Test group 2) vs. 6.30 ± 0.04 mmol/L (positive control): p = 0.0002]. Post-hoc statistical analysis using Tukey's multiple comparisons test revealed significant differences between normal saline and Test group 2 (p = 0.0102), normal saline and positive control (p = 0.0018), Test group 1 and Test group 2 (p = 0.0154) and, Test group 1 and positive control (p< 0.0028).

There were significant differences in the fasting blood glucose between the four experimental groups at the end of week 19: [6.47 ± 0.05 mmol/L (normal saline) vs. 6.50 ± 0.03 mmol/L (Test group 1) vs. 6.67 ± 0.05 mmol/L (Test group 2) vs. 6.69 ± 0.43 mmol/L (positive control): p = 0.0005]. Post-hoc statistical analysis using Tukey's multiple comparisons test revealed significant differences between normal saline and Test group 2 (p = 0.0079), normal saline and positive control (p = 0.0031), Test group 1 and Test group 2 (p = 0.2494) and, Test group 1 and positive control (p = 0.0124).

There were significant differences in the fasting blood glucose between the four experimental groups at the end of week 20: [6.42 ± 0.04 mmol/L (normal saline) vs. 6.43 ± 0.04 mmol/L (Test group 1) vs. 6.81 ± 0.07 mmol/L (Test group 2) vs. 6.82 ± 0.11 mmol/L (positive control): p = 0.0022]. Post-hoc statistical analysis using Tukey's multiple comparisons test revealed significant differences between normal saline and Test group 2 (p = 0.0021), normal saline and positive control (p = 0.0016), Test group 1 and Test group 2 (p = 0.0027) and, Test group 1 and positive control (p = 0.0021).

There were significant differences in the fasting blood glucose between the four experimental groups at the end of week 21: [6.61 ± 0.06 mmol/L (normal saline) vs. 6.58 ± 0.05 mmol/L (Test group 1) vs. 7.00 ± 0.04 mmol/L (Test group 2) vs. 7.15 ± 0.06 mmol/L (positive control): p< 0.0001]. Post-hoc statistical analysis using Tukey's multiple comparisons test revealed significant differences between normal saline and Test group 2 (p< 0.0001), normal saline and positive control (p< 0.0001), Test group 1 and Test group 2 (p< 0.0001) and, Test group 1 and positive control (p< 0.0001).

There were significant differences in the fasting blood glucose between the four experimental groups at the end of week 22: [6.77 ± 0.05 mmol/L (normal saline) vs. 6.72 ± 0.04 mmol/L (Test group 1) vs. 7.11 ± 0.05 mmol/L (Test group 2) vs. 7.20 ± 0.05 mmol/L (positive control): p< 0.0001]. Post-hoc statistical analysis using Tukey's multiple comparisons test revealed significant differences between normal saline and Test group 2 (p< 0.0001), normal saline and positive control (p< 0.0001), Test group 1 and Test group 2 (p< 0.0001) and, Test group 1 and positive control (p< 0.0001).

There were significant differences in the fasting blood glucose between the four experimental groups at the end of week 23: [6.88 ± 0.05 mmol/L (normal saline) vs. 6.90 ± 0.06 mmol/L (Test group 1) vs. 7.31 ± 0.03 mmol/L (Test group 2) vs. 7.36 ± 0.05 mmol/L (positive control): p< 0.0001]. Post-hoc statistical analysis using Tukey's multiple comparisons test revealed significant differences between normal saline and Test group 2 (p< 0.0001), normal saline and positive control (p< 0.0001), Test group 1 and Test group 2 (p< 0.0001) and, Test group 1 and positive control (p< 0.0001).

There were significant differences in the fasting blood glucose between the four experimental groups at the end of week 24: [6.94 ± 0.04 mmol/L (normal saline) vs. 7.42 ± 0.04 mmol/L (Test group 1) vs. 7.41 ± 0.05 mmol/L (Test group 2) vs. 7.45 ± 0.05 mmol/L (positive control):

p< 0.0001]. Post-hoc statistical analysis using Tukey's multiple comparisons test revealed significant differences between normal saline and Test group 2 (p< 0.0001), normal saline and positive control (p< 0.0001), Test group 1 and Test group 2 (p< 0.0001) and, Test group 1 and positive control (p< 0.0001).

*Low protein high calorie diet.* There were significant differences in the fasting blood glucose between the four experimental groups at the end of week 16: [6.50 ± 0.08 mmol/L (normal saline) vs. 6.60 ± 0.07 mmol/L (Test group 1) vs. 7.17 ± 0.06 mmol/L (Test group 2) vs. 7.04 ± 0.06 mmol/L (positive control): p< 0.0001]. Post-hoc statistical analysis using Tukey's multiple comparisons test revealed significant differences between normal saline and Test group 2 (p< 0.0001), normal saline and positive control (p< 0.0001), Test group 1 and Test group 2 (p< 0.0001) and, Test group 1 and positive control (p< 0.0001).

There were significant differences in the fasting blood glucose between the four experimental groups at the end of week 17: [6.83 ± 0.08 mmol/L (normal saline) vs. 6.93 ± 0.05 mmol/L (Test group 1) vs. 7.48 ± 0.04 mmol/L (Test group 2) vs. 7.44 ± 0.04 mmol/L (positive control): p< 0.0001]. Post-hoc statistical analysis using Tukey's multiple comparisons test revealed significant differences between normal saline and Test group 2 (p< 0.0001), normal saline and positive control (p< 0.0001), Test group 1 and Test group 2 (p< 0.0001) and, Test group 1 and positive control (p< 0.0001).

There were significant differences in the fasting blood glucose between the four experimental groups at the end of week 18: [7.23 ± 0.08 mmol/L (normal saline) vs. 7.30 ± 0.03 mmol/L (Test group 1) vs. 7.66 ± 0.05 mmol/L (Test group 2) vs. 7.69 ± 0.04 mmol/L (positive control): p< 0.0001]. Post-hoc statistical analysis using Tukey's multiple comparisons test revealed significant differences between normal saline and Test group 2 (p< 0.0001), normal saline and positive control (p< 0.0001), Test group 1 and Test group 2 (p< 0.0001) and, Test group 1 and positive control (p< 0.0001).

There were significant differences in the fasting blood glucose between the four experimental groups at the end of week 19: [7.45 ± 0.07 mmol/L (normal saline) vs. 7.49 ± 0.07 mmol/L (Test group 1) vs. 8.01 ± 0.05 mmol/L (Test group 2) vs. 8.03 ± 0.05 mmol/L (positive control): p< 0.0001]. Post-hoc statistical analysis using Tukey's multiple comparisons test revealed significant differences between normal saline and Test group 2 (p< 0.0001), normal saline and positive control (p< 0.0001), Test group 1 and Test group 2 (p< 0.0001) and, Test group 1 and positive control (p< 0.0001).

There were significant differences in the fasting blood glucose between the four experimental groups at the end of week 20: [7.70 ± 0.05 mmol/L (normal saline) vs. 7.56 ± 0.07 mmol/L (Test group 1) vs. 8.40 ± 0.06 mmol/L (Test group 2) vs. 8.36 ± 0.05 mmol/L (positive control): p< 0.0001]. Post-hoc statistical analysis using Tukey's multiple comparisons test revealed significant differences between normal saline and Test group 2 (p< 0.0001), normal saline and positive control (p< 0.0001), Test group 1 and Test group 2 (p< 0.0001) and, Test group 1 and positive control (p< 0.0001).

There were significant differences in the fasting blood glucose between the four experimental groups at the end of week 21: [8.08 ± 0.06 mmol/L (normal saline) vs. 8.56 ± 0.07 mmol/L (Test group 1) vs. 8.69 ± 0.06 mmol/L (Test group 2) vs. 8.74 ± 0.05 mmol/L (positive control): p< 0.0001]. Post-hoc statistical analysis using Tukey's multiple comparisons test revealed significant differences between normal saline and Test group 2 (p< 0.0001), normal saline and positive control (p< 0.0001), Test group 1 and Test group 2 (p< 0.0001) and, Test group 1 and positive control (p< 0.0001).

There were significant differences in the fasting blood glucose between the four experimental groups at the end of week 22: [8.36 ± 0.07 mmol/L (normal saline) vs. 8.35 ± 0.07 mmol/L (Test group 1) vs. 8.84 ± 0.07 mmol/L (Test group 2) vs. 8.91 ± 0.05 mmol/L (positive control):

p< 0.0001]. Post-hoc statistical analysis using Tukey's multiple comparisons test revealed significant differences between normal saline and Test group 2 (p< 0.0001), normal saline and positive control (p< 0.0001), Test group 1 and Test group 2 (p< 0.0001) and, Test group 1 and positive control (p< 0.0001).

There were significant differences in the fasting blood glucose between the four experimental groups at the end of week 23: [8.51 ± 0.07 mmol/L (normal saline) vs. 8.61 ± 0.04 mmol/L (Test group 1) vs. 9.01 ± 0.05 mmol/L (Test group 2) vs. 9.19 ± 0.07 mmol/L (positive control): p< 0.0001]. Post-hoc statistical analysis using Tukey's multiple comparisons test revealed significant differences between normal saline and Test group 2 (p< 0.0001), normal saline and positive control (p< 0.0001), Test group 1 and Test group 2 (p< 0.0001) and, Test group 1 and positive control (p< 0.0001).

There were significant differences in the fasting blood glucose between the four experimental groups at the end of week 24: [8.72 ± 0.06 mmol/L (normal saline) vs. 8.77 ± 0.04 mmol/L (Test group 1) vs. 9.29 ± 0.09 mmol/L (Test group 2) vs. 9.42 ± 0.07 mmol/L (positive control): p< 0.0001]. Post-hoc statistical analysis using Tukey's multiple comparisons test revealed significant differences between normal saline and Test group 2 (p< 0.0001), normal saline and positive control (p< 0.0001), Test group 1 and Test group 2 (p< 0.0001) and, Test group 1 and positive control (p< 0.0001). The graphical presentation of the mean fasting blood glucose at weekly interval during the treatment phase is shown in Fig 6.

**Oral glucose tolerance test at week 24 (Treatment phase).** *Standard diet.* There were no significant differences in the AUC values between the four experimental groups on week 24: [638.3 ± 2.88 mmol/L.min (normal saline) vs. 646.4 ± 2.10 mmol/L.min (Test group 1) vs. 642.5 ± 4.31 mmol/L.min (Test group 2) vs. 648 ± 3.96 mmol/L.min (positive control): p = 0.1645].

*Normal protein high calorie diet.* There were significant differences in the AUC values between the four experimental groups on week 24: [917.7 ± 2.99 mmol/L.min (normal saline) vs. 928.7 ± 3.17 mmol/L.min (Test group 1) vs. 1025 ± 4.90 mmol/L.min (Test group 2) vs. 1029 ± 6.18 mmol/L.min (positive control): p< 0.0001]. Post-hoc statistical analysis using Tukey's multiple comparisons test revealed significant differences between normal saline and Test group 1 (p< 0.0001), normal saline and positive control (p< 0.0001), Test group 1 and Test group 2 (p< 0.0001) and, Test group 1 and positive control (p< 0.0001).

*Low protein high calorie diet.* There were significant differences in the AUC values between the four experimental groups on week 24: [1120.7 ± 5.53 mmol/L.min (normal saline) vs. 1128.4 ± 5.39 mmol/L.min (Test group 1) vs. 1264 ± 8.98 mmol/L.min (Test group 2) vs. 1282 ± 6.07 mmol/L.min (positive control): p< 0.0001]. Post-hoc statistical analysis using Tukey's multiple comparisons test revealed significant differences between normal saline and Test group 1 (p< 0.0001), normal saline and positive control (p< 0.0001), Test group 1 and Test group 2 (p< 0.0001) and, Test group 1 and positive control (p< 0.0001). The graphical presentation of the mean blood glucose response and mean area under the curve during the treatment phase is shown in Fig 7.

**Serum lipids during the treatment phase.** *Standard diet.* There were no significant differences in total serum cholesterol between the four experimental groups on week 24: [1.67 ± 0.07 mmol/L (normal saline) vs. 1.60 ± 0.01 mmol/L (Test group 1) vs. 1.78 ± 0.04 mmol/L (Test group 2) vs. 1.77 ± 0.04 mmol/L (positive control): p = 0.1245].

There were no significant differences in serum triglycerides between the four experimental groups on week 24: [0.96 ± 0.05 mmol/L (normal saline) vs. 0.81 ± 0.05 mmol/L (Test group 1) vs. 0.99 ± 0.04 mmol/L (Test group 2) vs. 0.98 ± 0.06 mmol/L (positive control): p = 0.1983].

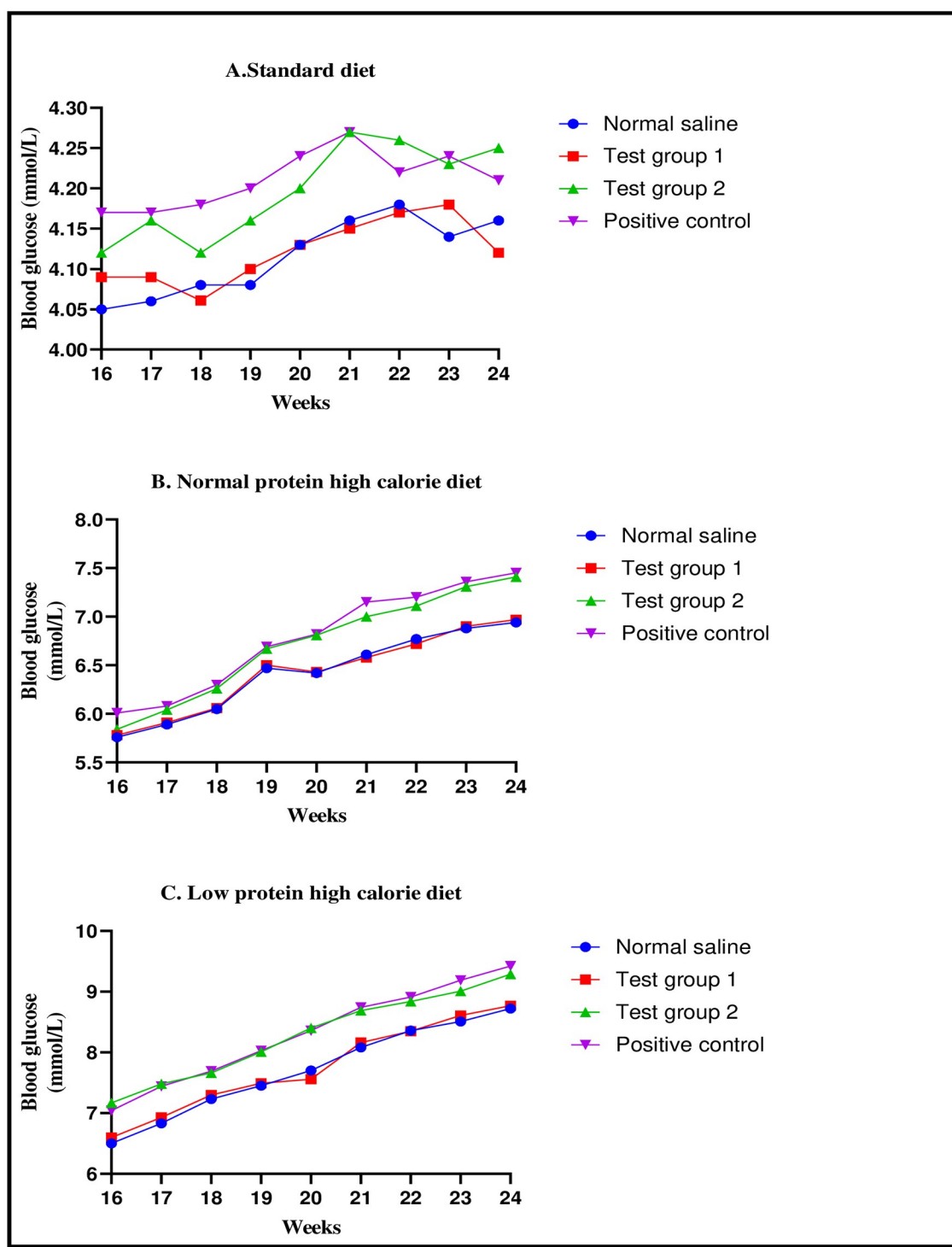

**Fig 6. Line graphs showing the mean fasting blood glucose (mmol/L) at weekly interval during the treatment phase.** Expressed as mean ± SEM. A (standard diet group), B (normal protein high calorie diet group), C (low protein high calorie diet group).

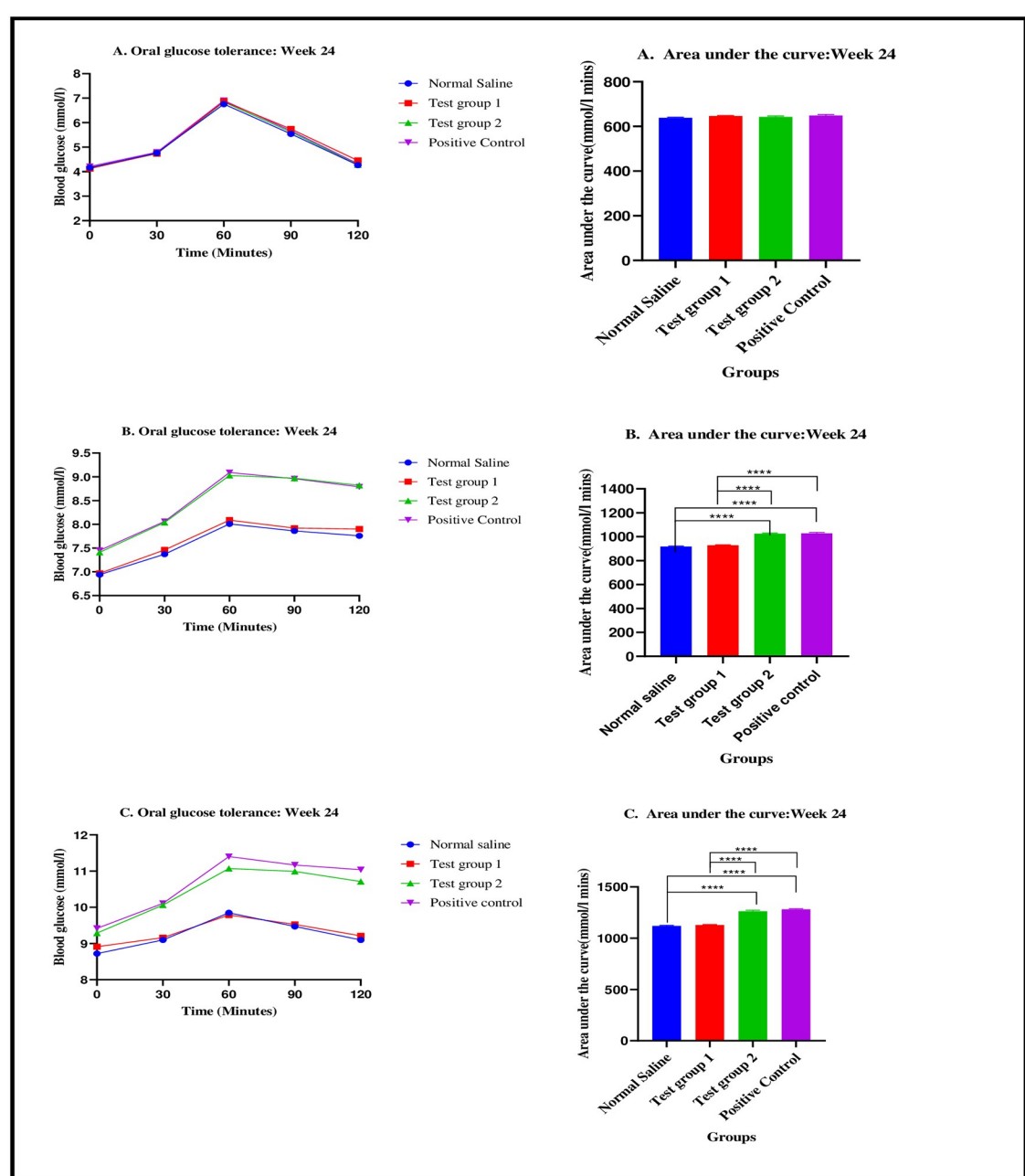

**Fig 7. Line graphs showing the mean blood glucose (mmol/l) response to an oral glucose bolus 2 g/kg over a 2-hour period and bar graphs using the mean area under the curve (mmol/l) during the oral glucose tolerance test.** Results are expressed as mean ± SEM. ****- p < 0.0001. A (Standard diet), B (Normal protein high calorie diet, C (Low protein high calorie diet).

There were no significant differences in LDL cholesterol between the four experimental groups on week 24: [0.49 ± 0.01 mmol/L (normal saline) vs. 0.45 ± 0.02 mmol/L (Test group 1) vs. 0.51 ± 0.02 mmol/L (Test group 2) vs. 0.50 ± 0.02 mmol/L (positive control): p = 0.1855].

There were no significant in differences HDL cholesterol between the four experimental groups on week 24: [0.73 ± 0.04 mmol/L (normal saline) vs. 0.73 ± 0.03 mmol/L (Test group 1) vs. 0.63 ± 0.03 mmol/L (Test group 2) vs. 0.64 ± 0.06 mmol/L (positive control): p = 0.1855].

The graphical presentation of the lipid profile results of the standard diet group during the treatment is shown in Fig 8.

*Normal protein high calorie diet*. There were significant differences in total serum cholesterol between the four experimental groups on week 24: [2.87 ± 0.08 mmol/L (normal saline) vs. 2.66 ± 0.06 mmol/L (Test group 1) vs. 6.05 ± 0.05 mmol/L (Test group 2) vs. 6.01 ± 0.05 mmol/L (positive control): p< 0.0001]. Post-hoc statistical analysis using Tukey's multiple comparisons test revealed significant differences between normal saline and Test group 2 (p< 0.0001), normal saline and positive control (p< 0.0001), Test group 1 and Test group 2 (p< 0.0001) and, Test group 1 and positive control (p< 0.0001).

There were significant differences in serum triglycerides between the four experimental groups on week 24: [3.14 ± 0.11 mmol/L (normal saline) vs. 2.85 ± 0.07 mmol/L (Test group 1) vs. 7.41 ± 0.13 mmol/L (Test group 2) vs. 7.49 ± 0.13 mmol/L (positive control): p< 0.0001]. Post-hoc statistical analysis using Tukey's multiple comparisons test revealed significant differences between normal saline and Test group 2 (p< 0.0001), normal saline and positive control (p< 0.0001), Test group 1 and Test group 2 (p< 0.0001) and, Test group 1 and positive control (p< 0.0001).

There were significant differences in LDL cholesterol between the four experimental groups on week 24: [1.38 ± 0.13 mmol/L (normal saline) vs. 1.32 ± 0.15 mmol/L (Test group 1) vs. 3.94 ± 0.03 mmol/L (Test group 2) vs. 3.83 ± 0.08 mmol/L (positive control): p< 0.0001]. Post-hoc statistical analysis using Tukey's multiple comparisons test revealed significant differences between normal saline and Test group 2 (p< 0.0001), normal saline and positive control (p< 0.0001), Test group 1 and Test group 2 (p< 0.0001) and, Test group 1 and positive control (p< 0.0001).

There were significant differences in HDL cholesterol between the four experimental groups on week 24: [2.02 ± 0.04 mmol/L (normal saline) vs. 1.99 ± 0.03 mmol/L (Test group 1) vs. 0.38 ± 0.05 mmol/L (Test group 2) vs. 0.40 ± 0.05 mmol/L (positive control): p< 0.0001]. Post-hoc statistical analysis using Tukey's multiple comparisons test revealed significant differences between normal saline and Test group 2 (p< 0.0001), normal saline and positive control (p< 0.0001), Test group 1 and Test group 2 (p< 0.0001) and, Test group 1 and positive control (p< 0.0001). The graphical presentation of the lipid profile results of the normal protein high calorie diet group during the treatment is shown in Fig 9.

*Low protein high calorie diet*. There were significant differences in total serum cholesterol between the four experimental groups on week 24: [3.77 ± 0.06 mmol/L (normal saline) vs. 3.64 ± 0.06 mmol/L (Test group 1) vs. 8.60 ± 0.15 mmol/L (Test group 2) vs. 8.65 ± 0.16 mmol/L (positive control): p< 0.0001]. Post-hoc statistical analysis using Tukey's multiple comparisons test revealed significant differences between normal saline and Test group 2 (p< 0.0001), normal saline and positive control (p< 0.0001), Test group 1 and Test group 2 (p< 0.0001) and, Test group 1 and positive control (p< 0.0001).

There were significant differences in serum triglycerides between the four experimental groups on week 24: [4.49 ± 0.14 mmol/L (normal saline) vs. 4.24 ± 0.12 mmol/L (Test group 1) vs. 9.25 ± 0.13 mmol/L (Test group 2) vs. 9.13 ± 0.18 mmol/L (positive control): p< 0.0001]. Post-hoc statistical analysis using Tukey's multiple comparisons test revealed significant differences between normal saline and Test group 2 (p< 0.0001), normal saline and positive control (p< 0.0001), Test group 1 and Test group 2 (p< 0.0001) and, Test group 1 and positive control (p< 0.0001).

There were significant differences in LDL cholesterol between the four experimental groups on week 24: [2.51 ± 0.11 mmol/L (normal saline) vs. 2.68 ± 0.10 mmol/L (Test group 1) vs. 4.78 ± 0.14 mmol/L (Test group 2) vs. 5.12 ± 0.09 mmol/L (positive control): p< 0.0001]. Post-hoc statistical analysis using Tukey's multiple comparisons test revealed significant differences

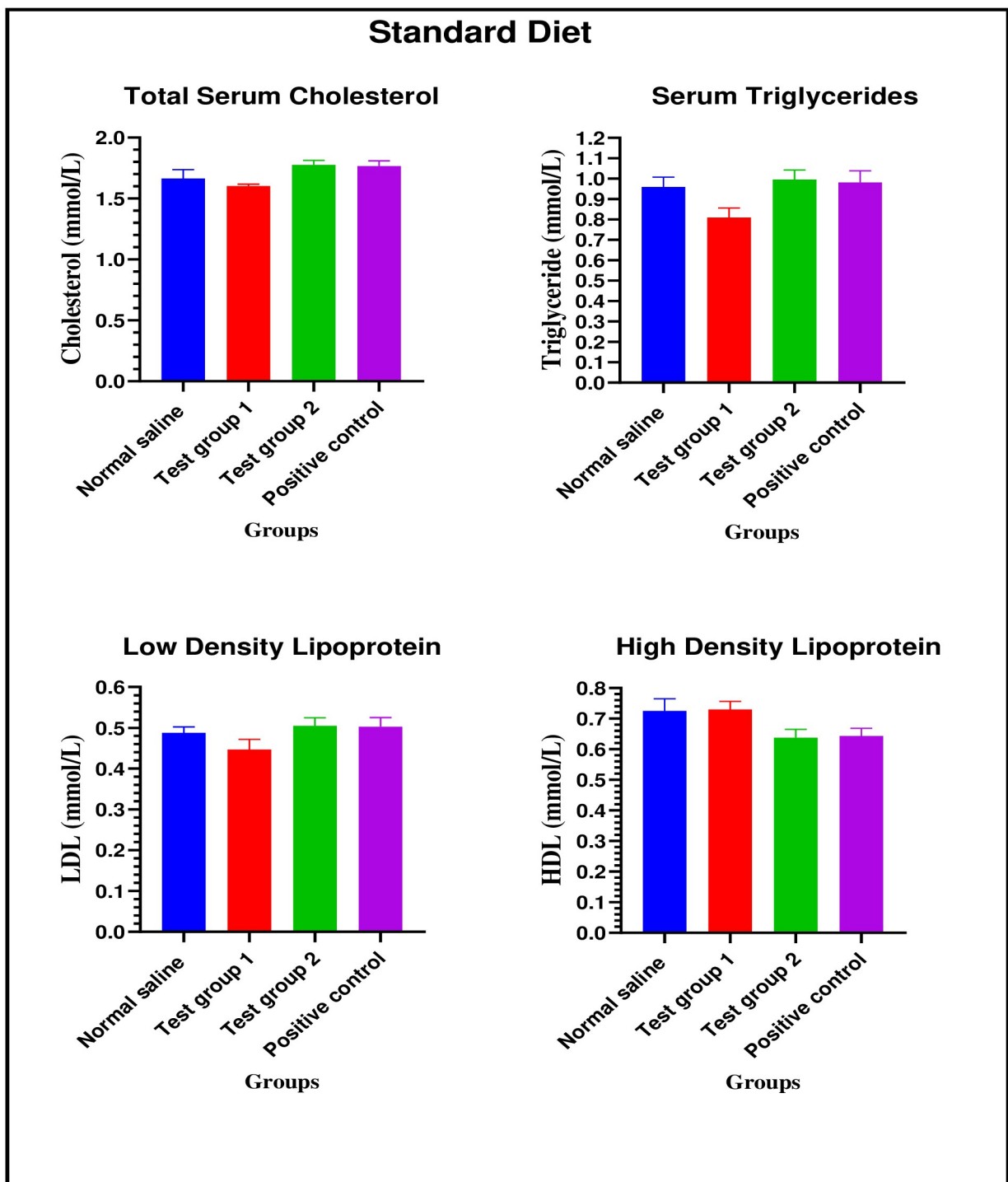

**Fig 8. Lipid profile of the standard diet group during the treatment phase.**

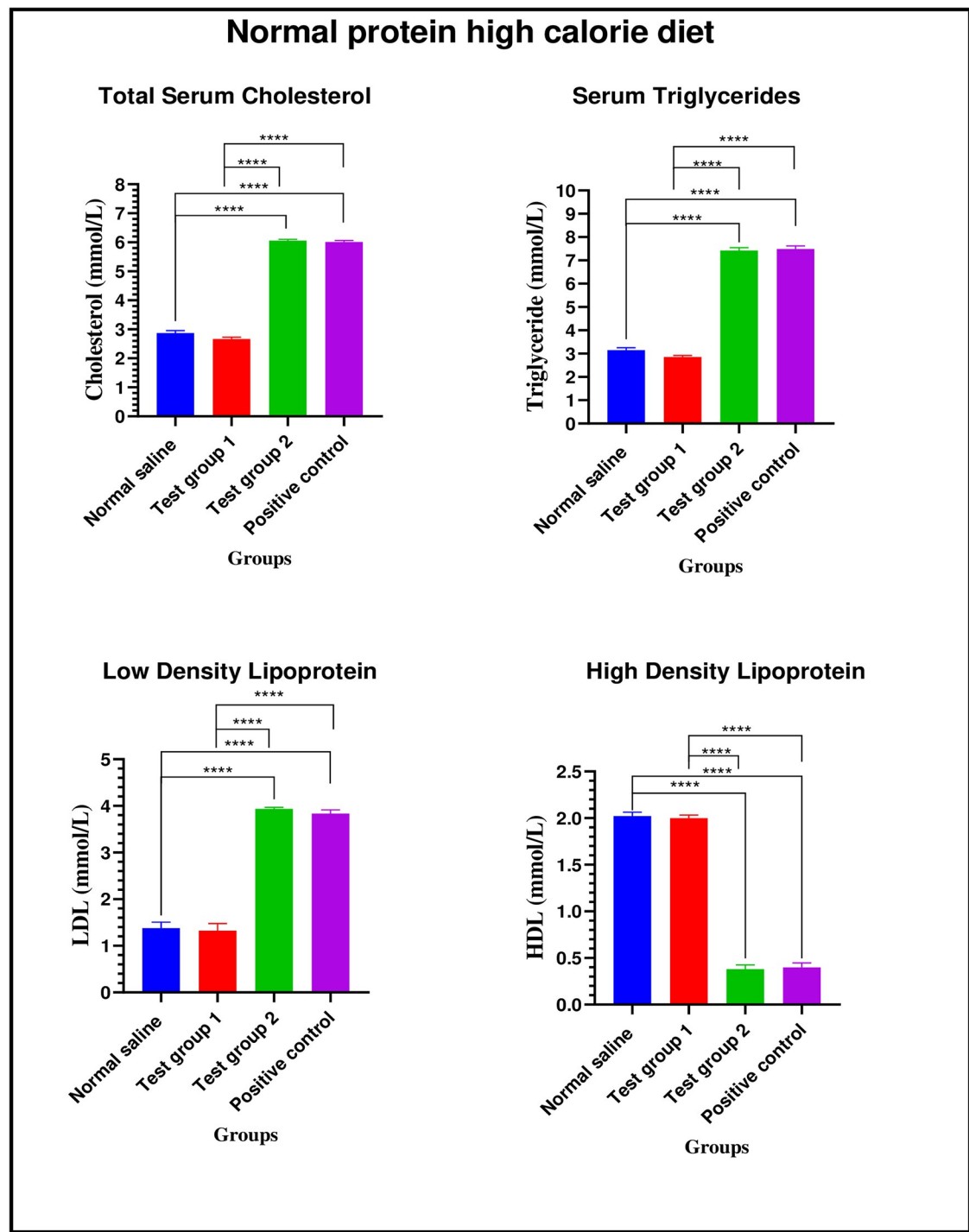

**Fig 9. Lipid profile of the normal protein high calorie diet group during the treatment phase.** Results are expressed as mean ± SEM. (****- p < 0.0001).

between normal saline and Test group 2 (p< 0.0001), normal saline and positive control (p< 0.0001), Test group 1 and Test group 2 (p< 0.0001) and, Test group 1 and positive control (p< 0.0001).

There were significant differences in HDL cholesterol between the four experimental groups on week 24: [3.37 ± 0.07 mmol/L (normal saline) vs. 3.37 ± 0.13 mmol/L (Test group 1) vs. 0.19 ± 0.03 mmol/L (Test group 2) vs. 0.20 ± 0.03 mmol/L (positive control): p< 0.0001]. Post-hoc statistical analysis using Tukey's multiple comparisons test revealed significant differences between normal saline and Test group 2 (p< 0.0001), normal saline and positive control (p< 0.0001), Test group 1 and Test group 2 (p< 0.0001) and, Test group 1 and positive control (p< 0.0001). The graphical presentation of the lipid profile results of the low protein high calorie diet group during the treatment is shown in Fig 10.

**Adipose tissue weight at the end of the treatment phase.** *Standard diet.* There were no significant differences in retroperitoneal adipose tissue weight between the four experimental groups [8.11 ± 0.06 g (normal saline) vs. 8.08 ± 0.04 g (Test group 1) vs. 8.38 ± 0.11 g (Test group 2) vs. 8.22 ± 0.11 g (positive control): p = 0.0980].

There were no significant differences in mesenteric adipose tissue weight between the four experimental groups [11.08 ± 0.08 g (normal saline) vs. 10.96 ± 0.12 g (Test group 1) vs. 11.24 ± 0.09 g (Test group 2) vs. 11.31 ± 0.08 g (positive control): p = 0.0748].

There were no significant differences in pericardial adipose tissue weight between the four experimental groups [2.82 ± 0.04 g (normal saline) vs. 2.76 ± 0.05 g (Test group 1) vs. 2.91 ± 0.04 g (Test group 2) vs. 2.92 ± 0.05 g (positive control): p = 0.0631]. The graphical presentation of the adipose tissue weights results of the standard diet group during the treatment is shown in Fig 11.

*Normal protein high calorie diet.* There were significant differences in retroperitoneal adipose tissue weight between the four experimental groups [10.49 ± 0.08 g (normal saline) vs. 10.25 ± 0.11 g (Test group 1) vs. 12.75 ± 0.09 g (Test group 2) vs. 12.82 ± 0.07 g (positive control): p< 0.0001]. Post-hoc statistical analysis using Tukey's multiple comparisons test revealed significant differences between normal saline and Test group 2 (p< 0.0001), normal saline and positive control (p< 0.0001), Test group 1 and Test group 2 (p< 0.0001) and, Test group 1 and positive control (p< 0.0001).

There were significant differences in mesenteric adipose tissue weight between the four experimental groups [13.64 ± 0.10 g (normal saline) vs. 13.54 ± 0.09 g (Test group 1) vs. 16.75 ± 0.11 g (Test group 2) vs. 16.60 ± 0.10 g (positive control): p< 0.0001]. Post-hoc statistical analysis using Tukey's multiple comparisons test revealed significant differences between normal saline and Test group 2 (p< 0.0001), normal saline and positive control (p< 0.0001), Test group 1 and Test group 2 (p< 0.0001) and, Test group 1 and positive control (p< 0.0001).

There were significant differences in pericardial adipose tissue weight between the four experimental groups [4.43 ± 0.78 g (normal saline) vs. 4.27 ± 0.07 g (Test group 1) vs. 8.35 ± 0.10 g (Test group 2) vs. 8.24 ± 0.73 g (positive control): p< 0.0001]. Post-hoc statistical analysis using Tukey's multiple comparisons test revealed significant differences between normal saline and Test group 2 (p< 0.0001), normal saline and positive control (p< 0.0001), Test group 1 and Test group 2 (p< 0.0001) and, Test group 1 and positive control (p< 0.0001). The graphical presentation of the adipose tissue weights results of the normal protein high calorie diet group during the treatment is shown in Fig 12.

*Low protein high calorie diet.* There were significant differences in retroperitoneal adipose tissue weight between the four experimental groups [14.46 ± 0.11 g (normal saline) vs. 14.52 ± 0.13 g (Test group 1) vs. 17.51 ± 0.09 g (Test group 2) vs. 17.48 ± 0.10 g (positive control): p< 0.0001]. Post-hoc statistical analysis using Tukey's multiple comparisons test revealed significant differences between normal saline and Test group 2 (p< 0.0001), normal saline and

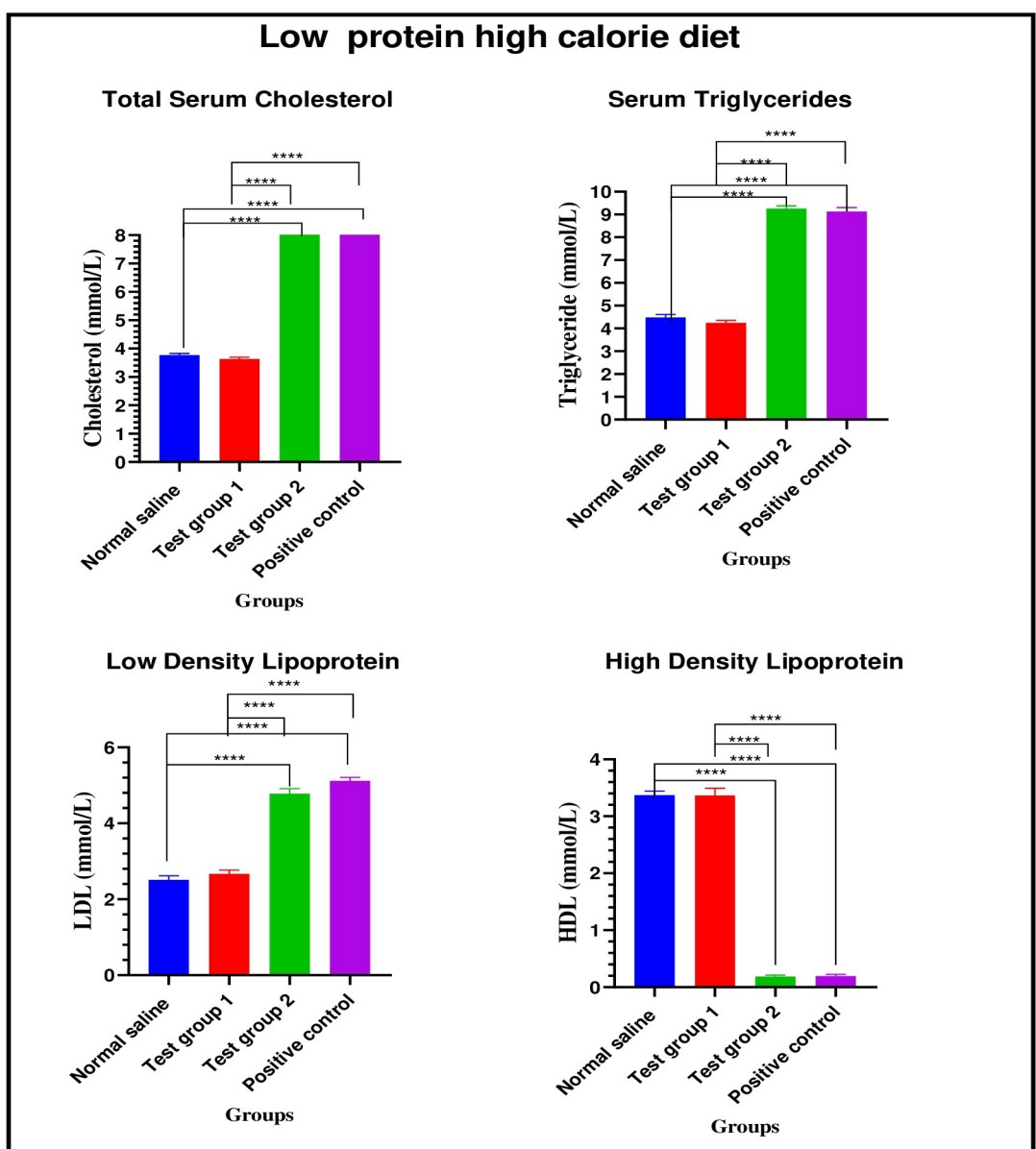

**Fig 10. Lipid profile of the low protein high calorie diet group during the treatment phase.** Results are expressed as mean ± SEM. (****- p < 0.0001).

positive control (p< 0.0001), Test group 1 and Test group 2 (p< 0.0001) and, Test group 1 and positive control (p< 0.0001).

There were significant differences in mesenteric adipose tissue weight between the four experimental groups [15.51 ± 0.12 g (normal saline) vs. 15.25 ± 0.07 g (Test group 1) vs. 20.47 ± 0.09 g (Test group 2) vs. 20.53 ± 0.89 g (positive control): p< 0.0001]. Post-hoc statistical analysis using Tukey's multiple comparisons test revealed significant differences between normal saline and Test group 2 (p< 0.0001), normal saline and positive control (p< 0.0001), Test group 1 and Test group 2 (p< 0.0001) and, Test group 1 and positive control (p< 0.0001).

## Standard diet

**Fig 11. Adipose tissue weights (grams) of the standard diet group during the treatment phase.**

There were significant differences in pericardial adipose tissue weight between the four experimental groups [6.43 ± 0.08 g (normal saline) vs. 6.21 ± 0.08 g (Test group 1) vs. 9.41 ± 0.10 g (Test group 2) vs. 9.46 ± 0.13 g (positive control): $p < 0.0001$]. Post-hoc statistical analysis using Tukey's multiple comparisons test revealed significant differences between normal saline and Test group 2 ($p < 0.0001$), normal saline and positive control ($p < 0.0001$), Test group 1 and Test group 2 ($p < 0.0001$) and, Test group 1 and positive control ($p < 0.0001$). The graphical presentation of the adipose tissue weights results of the normal protein high calorie diet group during the treatment is shown in Fig 13.

**Liver weights during the treatment phase.** *Standard diet.* There were no significant differences in mean liver weight between the four experimental groups [15.58 ± 0.32g (normal saline) vs. 16.04 ± 0.11 g (Test group 1) vs. 16.14 ±0.24 g (Test group 2) vs. 16.25 ± 0.14 g (positive control): $p = 0.2986$].

*Normal protein high calorie diet.* There were significant differences in mean liver weight between the four experimental groups [20.99 ± 0.25 g (normal saline) vs. 21.21 ± 0.29 g (Test group 1) vs. 23.34 ± 0.29 g (Test group 2) vs. 23.22 ± 0.19 g (positive control): $p < 0.0001$].

## Normal protein high calorie diet

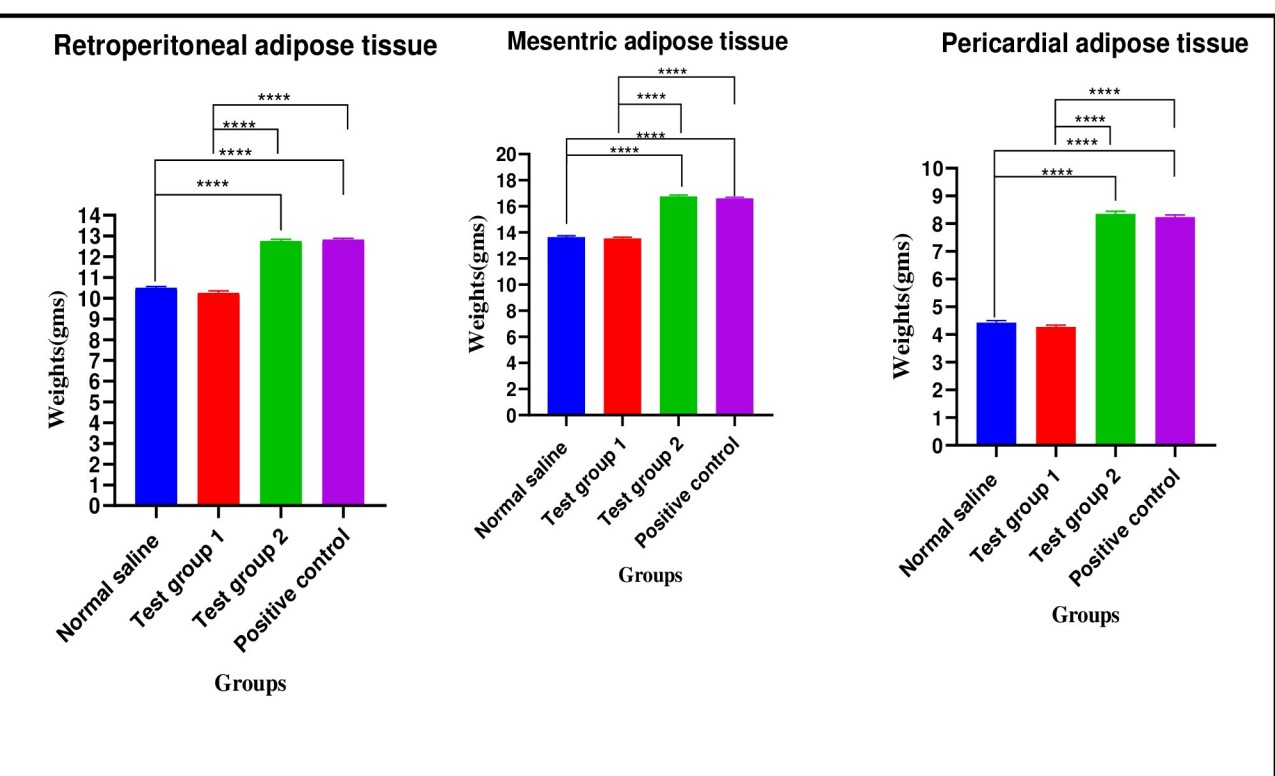

**Fig 12. Adipose tissue weights (grams) of the normal protein high calorie group during the treatment phase.**

Post-hoc statistical analysis using Tukey's multiple comparisons test revealed significant differences between normal saline and Test group 2 (p< 0.0001), normal saline and positive control (p< 0.0001), Test group 1 and Test group 2 (p< 0.0001) and, Test group 1 and positive control (p< 0.0001).

*Low protein high calorie diet.* There were significant differences in mean liver weight between the four experimental groups [24.92 ± 0.19 g (normal saline) vs. 25.11 ± 0.11 g (Test group 1) vs. 28.22 ± 0.18 g (Test group 2) vs. 28.72 ± 0.19 g (positive control): p< 0.0001]. Post-hoc statistical analysis using Tukey's multiple comparisons test revealed significant differences between normal saline and Test group 2 (p< 0.0001), normal saline and positive control (p< 0.0001), Test group 1 and Test group 2 (p< 0.0001) and, Test group 1 and positive control (p< 0.0001).

The graphical presentation of the liver weights during the treatment phase is shown in Fig 14.

**Fasting plasma insulin levels during the treatment phase.** *Standard diet.* There were no significant differences in fasting plasma insulin levels between the four experimental groups [4.420 ± 0.2529 mU/L (normal saline) vs. 4.750 ± 0.1668 mU/L (Test group 1) vs. 4.970 ± 0.1430 mU/L (Test group 2) vs. 4.910 ± 0.1501mU/L (positive control): p = 0.1641].

*Normal protein high calorie diet.* There were significant differences in fasting plasma insulin levels between the four experimental groups [5.750 ± 0.1759 mU/L (normal saline) vs. 5.590 ± 0.1882 mU/L (Test group 1) vs. 10.73 ± 0.2587 mU/L (Test group 2) vs. 11.02 ± 0.2555

# Low protein high calorie diet

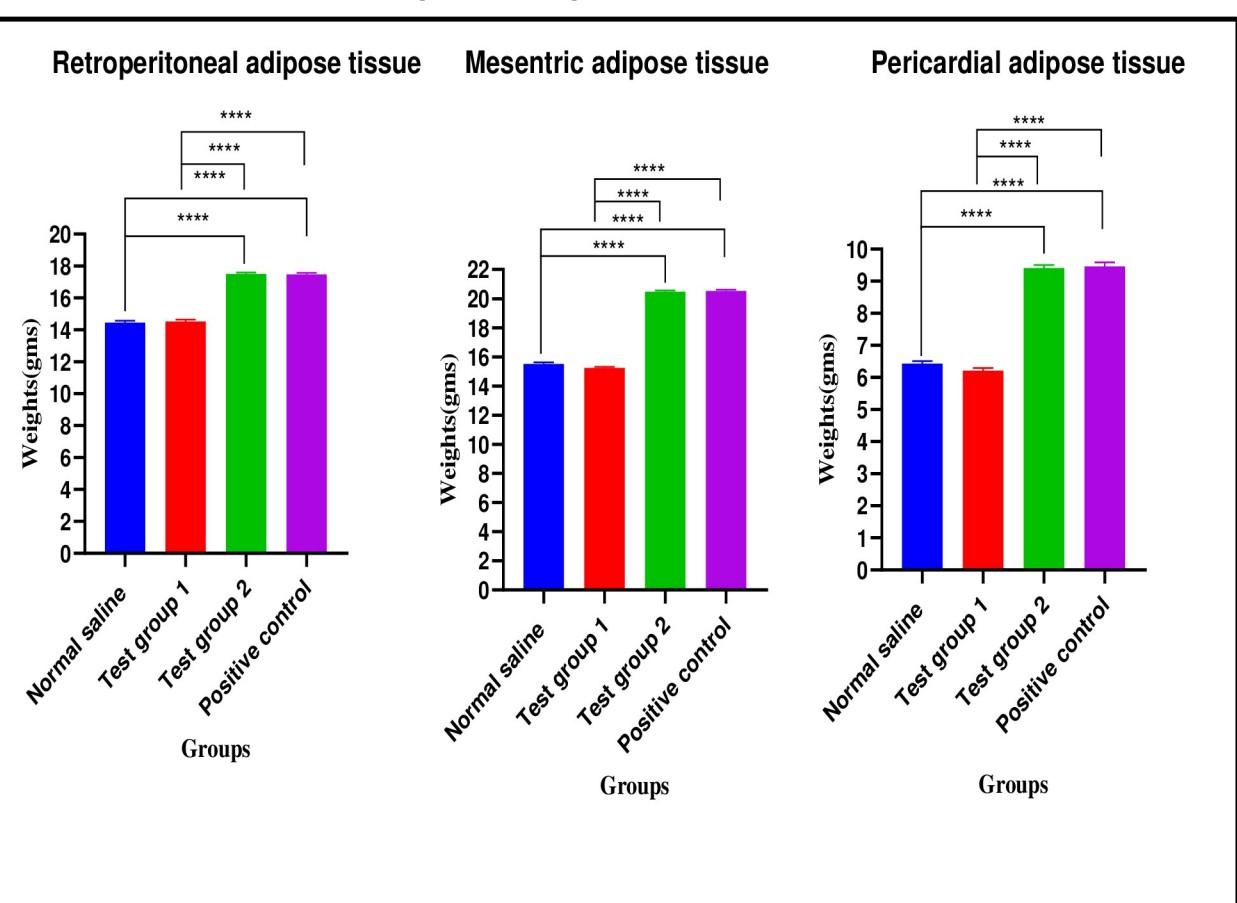

**Fig 13. Adipose tissue weights (grams) of the low protein high calorie group during the treatment phase.**

mU/L (positive control): p < 0.0001]. Post-hoc statistical analysis using Tukey's multiple comparisons test revealed significant differences between normal saline and Test group 2 (p< 0.0001), normal saline and positive control (p< 0.0001), Test group 1 and Test group 2 (p< 0.0001) and, Test group 1 and positive control (p< 0.0001).

*Low protein high calorie diet.* There were significant differences in fasting plasma insulin levels between the four experimental groups [8.410 ± 0.2238 mU/L (normal saline) vs. 8.840 ± 0.1956 mU/L (Test group 1) vs. 17.30 ± 0.2547 mU/L (Test group 2) vs. 18.04 ± 0.2837 mU/L (positive control): p < 0.0001]. Post-hoc statistical analysis using Tukey's multiple comparisons test revealed significant differences between normal saline and Test group 2 (p< 0.0001), normal saline and positive control (p< 0.0001), Test group 1 and Test group 2 (p< 0.0001) and, Test group 1 and positive control (p< 0.0001). The graphical presentation of the fasting serum insulin level during the treatment phase is shown in Fig 15.

**Homeostatic model assessment for insulin resistance (HOMA-IR) during the treatment phase.** *Standard diet.* There were no significant differences in HOMA-IR between the four experimental groups [0.8160 ± 0.05523 arbitrary units (normal saline) vs. 1.729 ± 0.05642 arbitrary units (Test group 1) vs. 3.535 ± 0.09142 arbitrary units (Test group 2) vs. 3.649 ± 0.08888 arbitrary units (positive control): p = 0.1173].

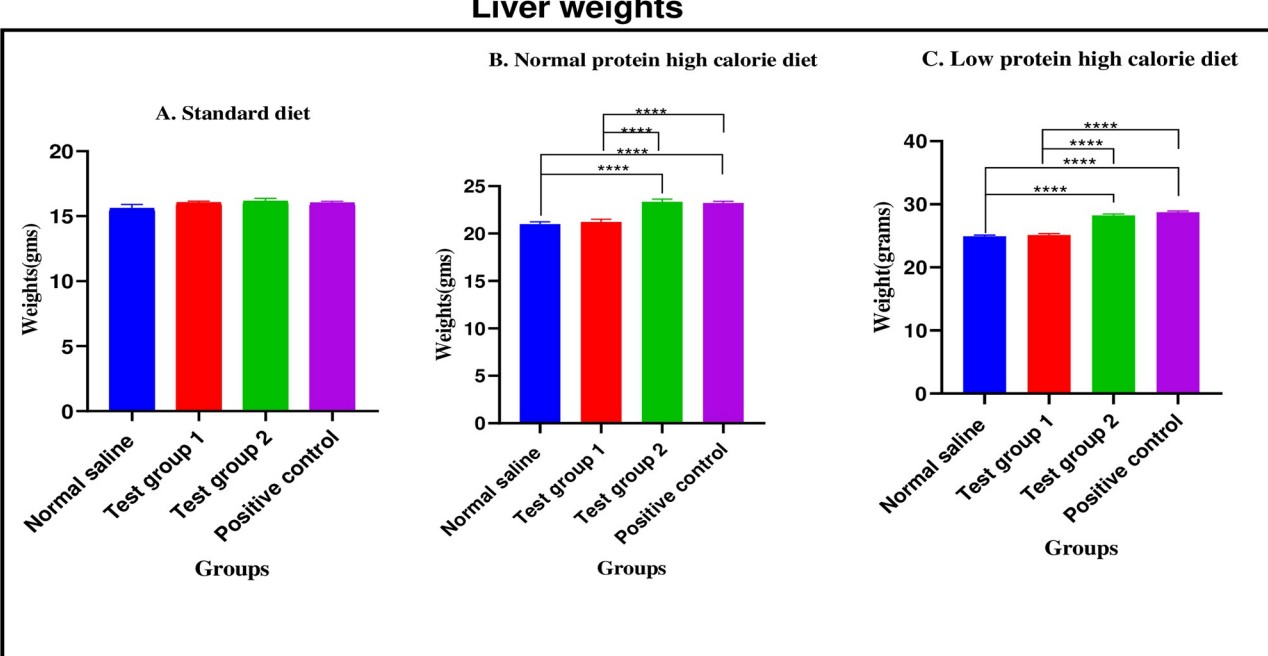

**Fig 14. Liver weights (grams) at the end of the treatment phase.** A (standard diet group), B (normal protein high calorie diet group), C (low protein high calorie diet group). Results are expressed as mean ± SEM. (****- p < 0.0001).

*Normal protein high calorie diet*. There were significant differences in HOMA-IR between the four experimental groups [1.772 ± 0.04624 arbitrary units (normal saline) vs. 0.8700 ± 0.03197 arbitrary units (Test group 1) vs. 0.9210 ± 0.02354 arbitrary units (Test group 2) vs. 0.9170 ± 0.02970 arbitrary units (positive control): <0.0001]. Post-hoc statistical analysis using Tukey's multiple comparisons test revealed significant differences between normal saline and Test group 2 (p< 0.0001), normal saline and positive control (p< 0.0001), Test group 1 and Test group 2 (p< 0.0001) and, Test group 1 and positive control (p< 0.0001).

*Low protein high calorie diet*. There were significant differences in HOMA-IR between the four experimental groups [3.264 ± 0.1052 arbitrary units (normal saline) vs. 3.446 ± 0.07845 arbitrary units (Test group 1) vs. 7.146 ± 0.1516 arbitrary units (Test group 2) vs. 7.552 ± 0.1189 arbitrary units (positive control): < 0.0001]. Post-hoc statistical analysis using Tukey's multiple comparisons test revealed significant differences between normal saline and Test group 2 (p< 0.0001), normal saline and positive control (p< 0.0001), Test group 1 and Test group 2 (p< 0.0001) and, Test group 1 and positive control (p< 0.0001). The graphical presentation of the HOMA-IR during the treatment phase is shown in Fig 16.

**Homeostatic model assessment for β-cell function (HOMA-B) during the treatment phase.** *Standard diet*. There were no significant differences in HOMA-B between the four experimental groups [17.78 ± 1.247 arbitrary units (normal saline) vs. 19.56 ± 0.7834 arbitrary units (Test group 1) vs. 20.38 ± 0.8163 arbitrary units (Test group 2) vs. 19.84 ± 0.7179 arbitrary units (positive control): p = 0.2274].

*Normal protein high calorie diet*. There were significant differences in HOMA-B between the four experimental groups [13.07± 0.5179 arbitrary units (normal saline) vs. 12.56 ± 0.6062 arbitrary units (Test group 1) vs. 25.47 ± 0.7097 arbitrary units (Test group 2) vs. 26.09 ± 0.6906 arbitrary units (positive control): <0.0001]. Post-hoc statistical

## Fasting serum insulin level

**Fig 15. Fasting plasma insulin levels (mU/L) at the end of the treatment phase.** A (standard diet group), B (normal protein high calorie diet group), C (low protein high calorie diet group). Results are expressed as mean ± SEM. (****- p < 0.0001).

analysis using Tukey's multiple comparisons test revealed significant differences between normal saline and Test group 2 (p< 0.0001), normal saline and positive control (p< 0.0001), Test group 1 and Test group 2 (p< 0.0001) and, Test group 1 and positive control (p< 0.0001).

*Low protein high calorie diet*. There were significant differences in HOMA-B between the four experimental groups [15.77± 0.4268 arbitrary units (normal saline) vs. 16.66 ± 0.4482 arbitrary units (Test group 1) vs. 33.75 ± 0.4470 arbitrary units (Test group 2) vs. 34.83 ± 0.7251 arbitrary units (positive control): <0.0001]. Post-hoc statistical analysis using Tukey's multiple comparisons test revealed significant differences between normal saline and Test group 2 (p< 0.0001), normal saline and positive control (p< 0.0001), Test group 1 and Test group 2 (p< 0.0001) and, Test group 1 and positive control (p< 0.0001). The graphical presentation of the HOMA-B during the treatment phase is shown in Fig 17.

**Hepatic triglycerides during the treatment phase.** *Standard diet*. There were no significant differences in hepatic triglycerides content between the four experimental groups [3.77 ± 0.12 mg/g (normal saline) vs. 3.75 ± 0.07 mg/g (Test group 1) vs. 3.95 ± 0.07 mg/g (Test group 2) vs. 3.96 ± 0.06 mg/g (positive control): p = 0.1664].

*Normal protein high calorie diet*. There were significant differences in hepatic triglycerides content between the four experimental groups [4.46 ± 0.15 mg/g (normal saline) vs. 4.80 ± 0.16 mg/g (Test group 1) vs. 6.15 ± 0.14 mg/g (Test group 2) vs. 6.05 ± 0.17 mg/g (positive control): p< 0.0001]. Post-hoc statistical analysis using Tukey's multiple comparisons test

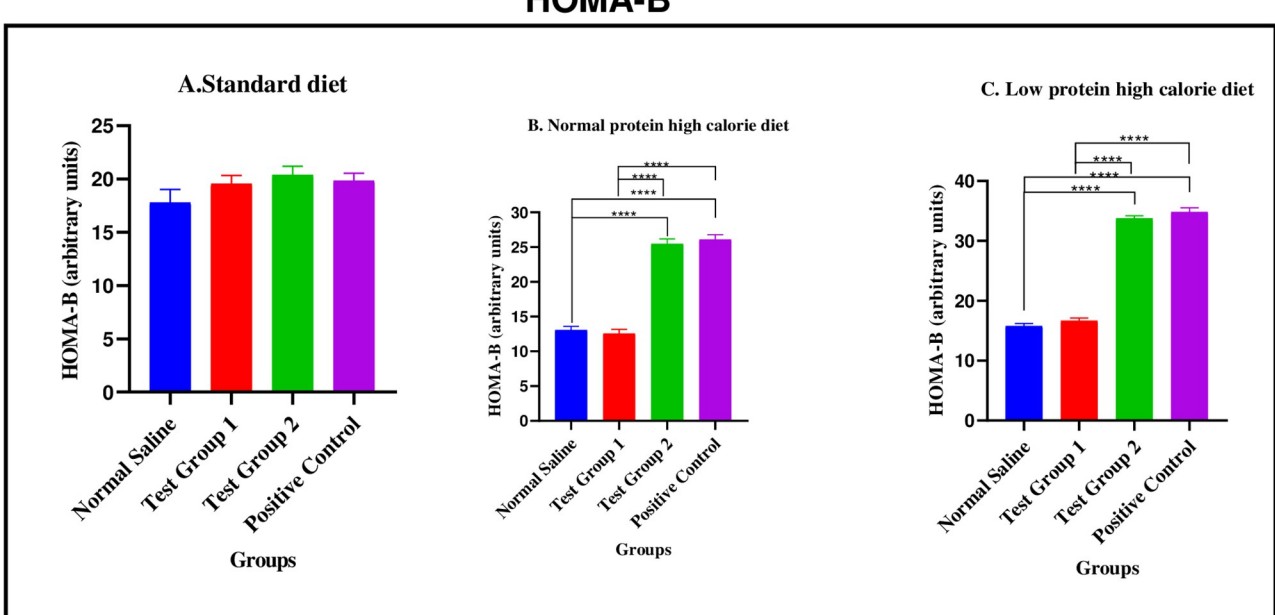

**Fig 16. HOMA-IR (arbitrary units) at the end of the treatment phase.** A (standard diet group), B (normal protein high calorie diet group), C (low protein high calorie diet group). Results are expressed as mean ± SEM. (****- p < 0.0001).

**Fig 17. HOMA-B (arbitrary units) at the end of the treatment phase.** A (standard diet group), B (normal protein high calorie diet group), C (low protein high calorie diet group). Results are expressed as mean ± SEM. (****- p < 0.0001).

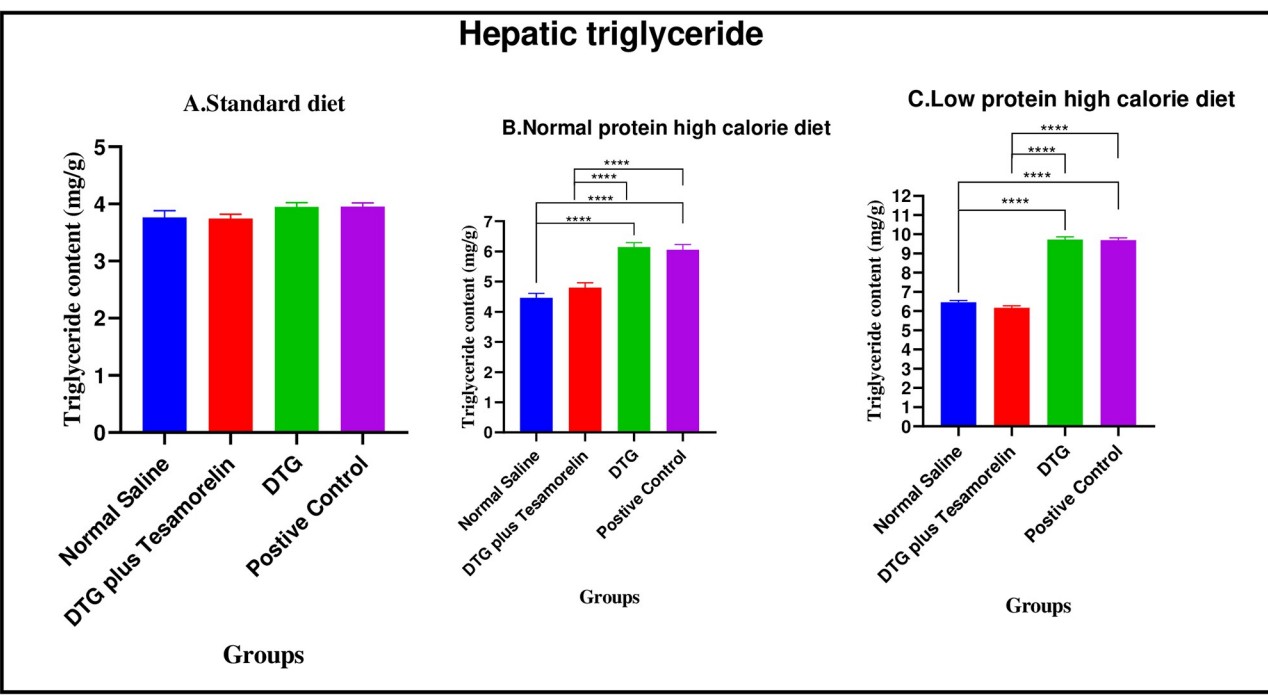

**Fig 18. Hepatic triglyceride (mg/g) at the end of the treatment phase.** Expressed as mean ± SEM. A (standard diet group), B (normal protein high calorie diet group), C (low protein high calorie diet group).

revealed significant differences between normal saline and Test group 2 (p< 0.0001), normal saline and positive control (p< 0.0001), Test group 1 and Test group 2 (p< 0.0001) and, Test group 1 and positive control (p< 0.0001).

*Low protein high calorie diet.* There were significant differences in hepatic triglycerides content between the four experimental groups [6.46 ± 0.10 mg/g (normal saline) vs. 6.18 ± 0.09 mg/g (Test group 1) vs. 9.73 ± 0.14 mg/g (Test group 2) vs. 9.70 ± 0.12 mg/g (positive control): p< 0.0001]. Post-hoc statistical analysis using Tukey's multiple comparisons test revealed significant differences between normal saline and Test group 2 (p< 0.0001), normal saline and positive control (p< 0.0001), Test group 1 and Test group 2 (p< 0.0001) and, Test group 1 and positive control (p< 0.0001). The graphical presentation of the hepatic triglycerides during the treatment phase is shown in Fig 18.

**Growth hormone levels during the treatment phase.** *Standard diet.* There were no significant differences in growth hormone levels between the four experimental groups [9.992 ± 0.2529 ng/mL (normal saline) vs. 10.45 ± 0.1840 ng/mL (Test group 1) vs. 9.923 ± 0.2186 ng/mL (Test group 2) vs. 9.921 ± 0.1418 ng/mL (positive control): p = 0.1344].

*Normal protein high calorie diet.* There were significant differences in growth hormone levels between the four experimental groups [9.728 ± 0.09886 ng/mL (normal saline) vs. 9.735 ± 0.1791 ng/mL (Test group 1) vs. 8.593 ± 0.2344 ng/mL (Test group 2) vs. 8.600 ± 0.1986 ng/mL (positive control): p = <0.0001]. Post-hoc statistical analysis using Tukey's multiple comparisons test revealed significant differences between normal saline and Test group 2 (p = 0.0006), normal saline and positive control (p = 0.0006), Test group 1 and Test group 2 (p = 0.0006) and, Test group 1 and positive control (p = 0.0006).

*Low protein high calorie diet.* There were significant differences in growth hormone levels between the four experimental groups [8.565 ± 0.1440 ng/mL (normal saline) vs.

8.654 ± 0.1497 ng/mL (Test group 1) vs. 7.529 ± 0.2114 ng/mL (Test group 2) vs.
7.514 ± 0.1063 ng/mL (positive control): p< 0.0001]. Post-hoc statistical analysis using
Tukey's multiple comparisons test revealed significant differences between normal saline
and Test group 2 (p = 0.0002), normal saline and positive control (p = 0.0002), Test group 1
and Test group 2 (p< 0.0001) and, Test group 1 and positive control (p< 0.0001). The
graphical presentation of growth hormone levels during the treatment phase is shown in Fig
19.

## Discussion

Although Sub-Saharan Africa contains only about 18% of the global population it bears a dis-
proportionate burden of the global HIV/AIDS burden with 60% of the people living with
HIV/AIDS (PLWH) being found there [11]. In addition, the majority of the world's poorest
people live in SSA meaning that many of the PLWH face nutritional challenges in terms of
accessing balanced diets bearing in mind that dietary protein is generally more expensive than
carbohydrates and lipids [12]. The objective of this study was to examine the obesogenic prop-
erties of the low protein high calorie diet as well as its interactions with the newly introduced
Integrase-based cART regimens. The normal protein high calorie and low protein high calorie
diets successfully induced; hyperglycemia, insulin resistance, and weight gain by the 15th week
of the study. The normal protein high calorie diet model has previously been shown to induce
metabolic syndrome in Sprague Dawley rats [13, 14].

The low protein high calorie diet was significantly obesogenic in this study and indeed dis-
played greater though non-significant obesogenic activity than the classical normal protein
high calorie diet (cafeteria diet). In addition, it had significantly more deleterious effects on

**Fig 19. Growth levels (ng/mL) at the end of the treatment phase.** A (standard diet group), B (normal protein high calorie diet group), C (low protein
high calorie diet group). Results are expressed as mean ± SEM. (****- p < 0.0001).

glucose tolerance than the cafeteria diet. There are several possible mechanistic explanations for the observed experimental findings.

Protein malnutrition is associated with inflammation due to intestinal dysbiotic microbiota (low diversity, increased prevalence of aerotolerant and decreased prevalence of beneficial commensal species) [15] and increased plasma concentrations of several mediators of the inflammatory cascade such as pro-inflammatory cytokines e.g., interleukin 6, C-reactive protein and the soluble receptors tumor necrosis factor α (sTNFR-p55 and sTNFR-p75) [16]. The antioxidant status is also significantly reduced [17]. Pro-inflammatory cytokines decrease insulin secretion in a clonal pancreatic β-cell line, explaining the classical features of metabolic syndrome, including central adiposity, hyperglycemia, and dyslipidemia seen in-vivo models [18]. An alternative theory is that protein malnutrition can cause the characteristics features of metabolic syndrome by inducing the production of hepatic fibroblast growth factor 21(FGF21) [19] which then interacts with β-klotho receptors in the brain triggering hyperphagia and ultimately resulting in energy overconsumption [20]. Similarly, a low protein diet is associated with histological evidence of hepatic steatosis that are attributed to endoplasmic reticulum stress, perturbation of autophagy, mitochondrial dysfunction, hepatocellular apoptosis, gut microbiota imbalance, dysregulation of microRNAs [21] and a loss of peroxisomes, which are important for normal liver metabolic function [22].

In cases of obesity, adipocytes undergo both hyperplasia and hypertrophy, displaying structural and functional deficiencies that ultimately alter their secretory and humoral characteristics [23]. The release of inflammatory agents by enlarged fat cells [24] is linked to the chronic low-grade inflammation (meta inflammation) characteristic of metabolic syndrome. Increased visceral fat is correlated with various adverse health effects. For instance, adipocytes laden with fat exhibit poor responsiveness to insulin stimulation [25] leading to hyperinsulinemia (insulin resistance) and hyperglycemia. This impaired insulin response extends beyond adipocytes, affecting other tissues such as skeletal muscle and the liver [26, 27].

The glucose handling in the animals belonging to the low protein high calorie diet group was significantly worse than in the animals belonging to the other dietary groups as shown by the fasting blood glucose and oral glucose tolerance tests. Both the fasting blood glucose and oral tolerance tests have been used as hallmark tests for the evaluation of insulin sensitivity and insulin resistance [28]. Previous published studies have shown that low protein diets are often associated with decreased glucose tolerance and reduced insulin secretion [29]. High calorie diets have been shown to have similar effects [30]. The morphology of the pancreatic islet has been shown to be altered in animal models of high calorie malnutrition [31]. The diminished number of β-cells per islet coupled with decreased levels of insulin secretion per unit β-cell may explain the observed hyperglycemia [32]. The increased inflammation and gut leakage may also explain the above experimental results [33].

The utilization of DTG is on the rise, particularly in low to middle-income countries, where it is incorporated into a single-tablet regimen known as tenofovir/lamivudine/dolutegravir (TLD). This is attributed to DTG's high resistance barrier, once-daily dosing, and its independence from pharmacologic boosting [34]. Consequently, it becomes crucial to assess potential adverse effects of DTG, especially among people living with HIV (PLWH), particularly those experiencing metabolic changes typically associated with older cART regimens. The second phase of this study aims to explore the interactions between DTG-based cART and diet, with a specific focus on a comparison with older cART regimens. In addition, it also aimed to investigate whether tesamorelin, a growth hormone secretion stimulant ameliorated the observed metabolic derangements.

There were significant weight gains in the animals receiving DTG-containing cART treatment regimens as well as those receiving the classical cART regimens in both the normal protein high calorie and low protein high calorie but not in the normal diet groups. These results are in line with those in literature where weight gain has been described as the most prominent metabolic side effect of second-generation integrase strand transfer inhibitors (INSTI) e.g., DTG [35]. The underlying pathophysiologic mechanisms underlying INSTI- associated weight gain remain unknown but proposed mechanisms include but are not limited to, direct impacts on adipogenesis [36], and gut microbiome disturbances [37] among others. It is noteworthy that the DTG effects on weight gain were prevented by co-administration with tesamorelin in this study, implying that it possesses additional activity against cART-induced obesity in addition to its documented anti lipodystrophy activity.

Several recent studies, predominantly conducted at single sites or with specific cohorts, indicate a higher incidence of weight gain in individuals initiating antiretroviral therapy (ART) with integrase strand transfer inhibitor (INSTI)-based regimens compared to those using protease inhibitors (PI) or non-nucleoside reverse transcriptase inhibitors (NNRTI)-based regimens. For instance, in a Brazilian cohort, people living with HIV (PLWH) on Raltegravir (RAL)-based regimens were seven times more likely to develop obesity (BMI $\geq$ 30 kg/m2) than those on NNRTI- or PI-based regimens [38]. Other observational studies also suggest that INSTI-based regimens, and particularly those utilizing dolutegravir (DTG) as part of ART, tend to be associated with a higher likelihood of weight gain [39]. In clinical trials, it has been observed that women and individuals of black ethnicity experience the most significant weight gain when using integrase inhibitors. Furthermore, there is evidence indicating that the nucleoside reverse transcriptase backbone may contribute to additional effects on weight gain, with tenofovir alafenamide potentially intensifying this effect [40].

Both the DTG-based and the classical cART regimens were associated with the development of hyperglycemia and impaired oral glucose tolerance when administered animals in the normal protein high calorie as well as the low protein high calorie diet groups respectively. These deleterious effects on glucose handling were however absent in the normal diet groups. In addition, these deleterious effects were prevented when the DTG-containing cART regimens were co-administered with tesamorelin.

Previous published studies have reported that integrase strand transfer inhibitor (INSTI) based cART regimens are associated with accelerated hyperglycemia in obese patient populations [40]. The pathophysiologic mechanisms responsible for the acceleration of hyperglycemia in these populations remain unclear but it has been postulated that these agents may contribute to beta-cell dysfunction and/or insulin resistance independent of weight again [41].

While the use of pharmacological doses of recombinant human Growth Hormone is known to lead to hyperglycemia, insulin resistance, fluid retention, and carpal tunnel syndrome [42], the utilization of analogs of growth hormone-releasing factor (GRF)/growth hormone-releasing hormone (GHRH) presents an alternative approach. These analogs stimulate natural increases in growth hormone (GH) levels while preserving the negative feedback mechanisms of insulin-like growth factor-1 (IGF-1). This helps address the metabolic irregularities and changes in body composition associated with low GH levels particularly with regard to hyperglycemia [43]. Indeed, MR-409, a GHRH receptor agonist, has been shown to induce Akt signaling via activation of insulin receptor substrate 2 (IRS2), a central controller of survival and growth in β-cells, occurs through a PKA-dependent mechanism [44]. The elevation in the functionality of the cAMP/PKA/CREB/IRS2 pathway induced by MR-409 was linked to a reduction in β-cell mortality and enhanced insulin secretion in both mouse and human islets exposed to proinflammatory cytokines. In addition, type 1 diabetic mice treated with MR-409 which (induced via the administration of low-dose streptozotocin) demonstrated

improved glucose regulation, elevated insulin levels, and maintenance of β-cell mass [45]. The foregoing discussion therefore provides a probable explanation for the improved glycemic control observed when tesamorelin was co-administered with the various cART regimens in this study. Both the DTG-containing and classical cART regimens induced the central adiposity, dyslipidemia (hypertriglyceridemia, elevated LDL and total cholesterol, lowered HDL-cholesterol), as well as the non-alcoholic fatty liver disease (as shown by the elevated hepatic index and hepatic triglycerides) that are characteristic of cART associated metabolic dysfunction. It is noteworthy that this cART associated metabolic dysfunction was absent in the normal diet groups.

Integrase strand transfer inhibitor (INSTI) based antiretroviral therapy has been reported to exhibit minimal increases in total cholesterol, serum triglyceride levels, LDL-cholesterol as well as causing an increase in HDL-cholesterol in a general population [46]. These results were replicated in this study in the normal diet group. However, dyslipidemia was observed in both the normal protein high calorie and low protein high calorie groups indicating that the DRG-based cART regimen interacts with high calorie diets in a manner analogous to that of the traditional cART. This to our knowledge, is the first study that has attempted to investigate the interaction between INSTI-based cART regimen and a high calorie diet and is a potentially novel finding.

Initiation of cART is often associated with central adiposity and eventually an increase in body weight [47]. Central adiposity is associated with low serum growth hormone levels [48]. Indeed, GH serum levels have been reported to be significantly lower in persons with lipodystrophy regardless of HIV status with the extent of visceral adipose tissue accumulation being closely correlated to the level of blunting of GH secretion [49]. Additionally, in-depth physiological investigations have revealed a diminished growth hormone (GH) secretion per pulse in individuals with normal pulse frequency. GH plays a role in increasing lipolysis and inhibiting de novo lipogenesis, establishing a mechanistic connection between decreased GH secretion and the documented buildup of abdominal fat and hepatic steatosis within this specific patient group [50]. The foregoing discussion provides an explanation for the ameliorative effects of tesamorelin when co-administered together with cART in both the low protein high calorie and normal protein diets in this study.

In particular, tesamorelin significantly ameliorated the central adiposity, possessed significant antidyslipidemic effects, and reduced hepatic adiposity. These beneficial effects mirror those in published literature where it has been reported to significantly reduce visceral fat by nearly 20%, decrease triglycerides by roughly 15%, reduced liver fat thereby halting the progression of hepatic fibrosis in patients with HIV-associated non-alcoholic fatty liver disease (NAFLD). This approach is accompanied by positive alterations in hepatic gene expression, contributing to an enhanced quality of life for patients. [51]. The limitation of this study is that the rats were not HIV positive and therefore cannot completely reproduce the cART dysregulations seen in HIV positive human subject.

## Conclusion

This study showed that the low protein high calorie diet was obesogenic. These obesogenic activities were as great as /exceeded that of the classical cafeteria diet (normal protein high calorie diet) in addition this lowprotein high calorie diet interacted with both IICR and classical cART drug regimen to reproduce cART associated metabolic dysregulations. These dysregulations were however reversed by co-treatment with tesamorelin indicating the possible involvement of the growth hormone system dysfunction in its pathophysiology.

The finding from this study therefore may provide a potential pathophysiologic explanation for the observed increased mortality rate seen in sub-Saharan Africa when cART is initiated in patients with nutritional insufficiency.

## Supporting information

**S1 File. Mean weekly body weight during diet induction phase.**
(PDF)

**S2 File. Mean fasting body glucose levels during the diet induction phase.**
(PDF)

**S3 File. Oral Glucose tolerance at week 15.**
(PDF)

**S4 File. Area under the curve at week 15.**
(PDF)

**S5 File. Mean weekly body weight for standard diet group during treatment phase.**
(PDF)

**S6 File. Mean weekly body weight for NPHC diet group during treatment phase.**
(PDF)

**S7 File. Mean weekly body weight for LPHC diet group during treatment phase.**
(PDF)

**S8 File. Mean weekly fasting blood glucose levels for standard diet group during treatment phase.**
(PDF)

**S9 File. Mean weekly fasting blood glucose levels for NPHC group during treatment phase.**
(PDF)

**S10 File. Mean weekly fasting blood glucose levels for LPHC diet group during treatment phase.**
(PDF)

**S11 File. Oral glucose tolerance test for standard diet group during treatment phase.**
(PDF)

**S12 File. Oral glucose test for NPHC diet group during treatment phase.**
(PDF)

**S13 File. Oral glucose tolerance test for LPHC diet group during treatment phase.**
(PDF)

**S14 File. Area under the standard diet during the oral glucose tolerance test during treatment phase.**
(PDF)

**S15 File. Area under the curve for LPHC during the oral glucose tolerance test during treatment phase.**
(PDF)

**S16 File. Area under the curve for NPHC during the oral glucose tolerance test during treatment phase.**
(PDF)

**S17 File. Serum triglyceride for the standard diet during the treatment phase.**
(PDF)

**S18 File. LDL for the standard diet during the treatment phase.**
(PDF)

**S19 File. HDL for the standard diet during the treatment phase.**
(PDF)

**S20 File. Total serum cholesterol for the NPHC during the treatment phase.**
(PDF)

**S21 File. Serum triglyceride for the NPHC during the treatment phase.**
(PDF)

**S22 File. Total serum cholesterol for the standard diet during the treatment phase.**
(PDF)

**S23 File. LDL for the NPHC during the treatment phase.**
(PDF)

**S24 File. HDL for the NPHC during the treatment phase.**
(PDF)

**S25 File. Serum triglyceride for the LPHC during the treatment phase.**
(PDF)

**S26 File. LDL for the LPHC during the treatment phase.**
(PDF)

**S27 File. HDL for the LPHC during the treatment phase.**
(PDF)

**S28 File. Retroperiotoneal adipose tissue for the standard diet group during the treatment phase.**
(PDF)

**S29 File. Mesenteric adipose tissue for the standard diet group during the treatment phase.**
(PDF)

**S30 File. Pericardial adipose tissue for the standard diet group during the treatment phase.**
(PDF)

**S31 File. Retroperiotoneal adipose tissue for the NPHC diet during the treatment phase.**
(PDF)

**S32 File. Mesenteric adipose tissue for the NPHC diet during the treatment phase.**
(PDF)

**S33 File. Pericardial adipose tissue for the NPHC during the treatment phase.**
(PDF)

**S34 File. Retroperiotoneal adipose tissue for the LPHC diet during the treatment phase.**
(PDF)

**S35 File. Mesenteric adipose tissue for the LPHC diet during the treatment phase.**
(PDF)

**S36 File. Pericardial adipose tissue for the LPHC during the treatment phase.**
(PDF)

**S37 File. Total cholesterol for the LPHC during the treatment phase.**
(PDF)

**S38 File. Liver weights for the LPHC diet group during the treatment phase.**
(PDF)

**S39 File. HOMA IR for the standard diet group during the treatment phase.**
(PDF)

**S40 File. Liver weights for the NPHC diet group during the treatment phase.**
(PDF)

**S41 File. HOMA IR for the LPHC diet group during the treatment phase.**
(PDF)

**S42 File. HOMA IR for the NPHC diet group during the treatment phase.**
(PDF)

**S43 File. Liver weights for the standard diet group during the treatment phase.**
(PDF)

**S44 File. Fasting insulin levels for the NPHC diet group during the treatment phase.**
(PDF)

**S45 File. Fasting insulin levels for the LPHC diet group during the treatment phase.**
(PDF)

**S46 File. Fasting insulin levels for the standard diet group during the treatment phase.**
(PDF)

**S47 File. HOMA B for the LPHC diet group during the treatment phase.**
(PDF)

**S48 File. Hepatic triglyceride for the standard diet group during the treatment phase.**
(PDF)

**S49 File. Hepatic triglyceride for the NPHC diet group during the treatment phase.**
(PDF)

**S50 File. Hepatic triglyceride for the LPHC diet group during the treatment phase.**
(PDF)

**S51 File. Growth hormone levels for the standard diet group during the treatment phase.**
(PDF)

**S52 File. Growth hormone levels for the NPHC diet group during the treatment phase.**
(PDF)

**S53 File. Growth hormone levels for the LPHC diet group during the treatment phase.**
(PDF)

**S54 File. HOMA B for the NPHC diet group during the treatment phase.**
(PDF)

**S55 File. HOMA B for the standard diet group during the treatment phase.**
(PDF)

## Acknowledgments

The authors want to express their gratitude for the valuable technical support provided by Ms. Abigeal Mutua, Mr. Horo Mwaura, Martin Omondi and Preetesh Jakhari.

## Author Contributions

**Conceptualization:** Boniface M. Chege, Peter W. Mwangi, Charles G. Githinji, Frederick Bukachi.

**Data curation:** Boniface M. Chege, Charles G. Githinji.

**Formal analysis:** Boniface M. Chege, Charles G. Githinji, Frederick Bukachi.

**Investigation:** Boniface M. Chege, Peter W. Mwangi, Charles G. Githinji, Frederick Bukachi.

**Methodology:** Boniface M. Chege, Peter W. Mwangi, Charles G. Githinji, Frederick Bukachi.

**Supervision:** Peter W. Mwangi, Charles G. Githinji, Frederick Bukachi.

**Validation:** Boniface M. Chege, Frederick Bukachi.

**Writing – original draft:** Boniface M. Chege.

**Writing – review & editing:** Boniface M. Chege, Peter W. Mwangi, Charles G. Githinji, Frederick Bukachi.

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
