## [Decision Letter · Decision Letter 0]

3 Dec 2023

PONE-D-23-33547Dietary regimens appear to possess significant effects on the development of Combined Antiretroviral Therapy (cART)-associated metabolic syndromePLOS ONE

Dear Dr. CHEGE,

Thank you for submitting your manuscript to PLOS ONE. After careful consideration, we feel that it has merit but does not fully meet PLOS ONE’s publication criteria as it currently stands. Therefore, we invite you to submit a revised version of the manuscript that addresses the points raised during the review process.

The reviewers are impressed with the manuscript. However, they have a few suggestions for improvement that the authors need to consider in revising the manuscript.

We look forward to receiving your revised manuscript.

Kind regards,

Chika Kingsley Onwuamah, Ph.D.

Academic Editor

PLOS ONE

- https://doi.org/10.1007/s11904-020-00483-5

- https://doi.org/10.1371/journal.pone.0179538

In your revision ensure you cite all your sources (including your own works), and quote or rephrase any duplicated text outside the methods section. Further consideration is dependent on these concerns being addressed.

Reviewers' comments:

Reviewer's Responses to Questions

**Comments to the Author**

1. Is the manuscript technically sound, and do the data support the conclusions?

Reviewer #1: Yes

Reviewer #2: Yes

Reviewer #3: Yes

2. Has the statistical analysis been performed appropriately and rigorously? 

Reviewer #1: Yes

Reviewer #2: Yes

Reviewer #3: Yes

3. Have the authors made all data underlying the findings in their manuscript fully available?

Reviewer #1: Yes

Reviewer #2: Yes

Reviewer #3: Yes

4. Is the manuscript presented in an intelligible fashion and written in standard English?

Reviewer #1: Yes

Reviewer #2: Yes

Reviewer #3: Yes

5. Review Comments to the Author

Reviewer #1: This study investigated the interactions between a low protein high calorie (LPHC) diet and an integrase inhibitor- containing antiretroviral drug regimen in light of evidence suggesting that the initiation of cART in patients with poor

nutritional status is a predictor of mortality independent of immune status.

The manuscript is technically sound and the results and data supports the conclusions.

The authors have made a suitable case on obesogenic effects of low protein high calorie diets with combined antiretrovial therapy.

The statistical analysis was adequately done, with appropraite statistical methods emplyed to differenciate the data.

The authors have made obtained data underlying their findings fully available as part of the manuscript.

The manuscript was presented in an understanding, simple intelligible fashion and written in fairly standard English.

Reviewer #2: A well written manuscript, but the abstract had no results. The section of the abstract titled "Methods and Results" only had methods. There was then a conclusion without having provided study results. I found no other significant issues requiring corrections.

Reviewer #3: Review of Dietary regimens appear to possess significant effects on the development of

Combined Antiretroviral Therapy (cART)-associated metabolic syndrome

This manuscript speaks on a study carried out to show the role/effect of dietary regimes in antiretroviral therapy leading to cART -associated metabolic syndrome and the potential of dietary regimens to increase chances of combined antiretroviral therapy -associated metabolic syndrome, bringing to fore, a need for dietary considerations in drug administration. Though this study draws attention to need for dietary consideration in drug administration. Generally more work needs to be done for better understanding of work done, ensure clarity and easy navigation and reference of documented information.

General comments

Integrase inhibitor-containing regimens should be abbreviated for easier read. A lot of your abbreviations were not given its full meaning at first mention.. (PLWH, NNRTI, INSTI, DTG, NAFLD etc). Ensure this is written in full for the sake of better understanding of concepts.

The authors would also need to be consistent with your abbreviations, "PLWH' should not be interchanged with "PWH"

Reviewer's comment

1. I suggest the experimental design for administering different diets (especially for phase 2) to the experimental animals be represented in tabular form for easy reference (Line 101-114)

2.The reporting of significant changes in parameters weekly, after which a table to show the same information makes it quite bulky to read. I believe a summary stating the point of significance and the amount of weeks for which it was significant should suffice. While the Table can show the full information.

3.“On”? Was the FBG recorded on the Call® 123 EZ II or it was an instrument/equipment used to record? If the latter is the case then “via” should be used (Line 122)

4.The AUC- What curve is being referred to? Curve of which graph? Where was it mentioned that a graph was plotted or was it stated in the cited reference. (line 131-132).

5.The calculation (Line 139-140) looks like a screenshot cropped and pasted in the document. I suggest the calculation be re-input manually

6.Generally, the use of abbreviations needs to be worked on. The essence of using abbreviations is to help simplify continuous use of some compound terms, for clarity sake. It is noticed that some abbreviations are not defined at first mention (e.g PWLH), some are not used consistently (PWLH/PWH), and some have been previously mentioned but still get defined and mentioned concurrently ( e.g AUC. see line 351 for a term that has been defined previously see line 131).

Discussion: line 1010,I believe the population of sub Saharan Africa is more than 6% of the world's population, ensure your findings are made correctly. 7.Please fact check and add reference.

6. PLOS authors have the option to publish the peer review history of their article (what does this mean?). If published, this will include your full peer review and any attached files.

Reviewer #1: No

Reviewer #2: **Yes: **Ebiere Clara HERBERTSON

Reviewer #3: **Yes: **Olufemi Samuel AMOO

---

## [Author Response · Author response to Decision Letter 0]

3 Jan 2024

Editor and Reviewer comments:

1.Editor: Please ensure that your manuscript meets PLOS ONE's style requirements, including those for file naming. The PLOS ONE style templates can be found at

Authors response: The authors appreciate the reviewer’s comment and manuscript has been revised to meet PLOS ONE requirement,

2. Editor: We noticed you have some minor occurrence of overlapping text with the following previous publication(s), which needs to be addressed:

- https://doi.org/10.1007/s11904-020-00483-5

- https://doi.org/10.1371/journal.pone.0179538

Authors response: The authors agree with editor on overlapping text of the two previous publications. The authors have rephrased the paragraphs affected to correct the overlap.

 3. Editor: In your Data Availability statement, you have not specified where the minimal data set underlying the results described in your manuscript can be found. PLOS defines a study's minimal data set as the underlying data used to reach the conclusions drawn in the manuscript and any additional data required to replicate the reported study findings in their entirety. All PLOS journals require that the minimal data set be made fully available. 

Authors response: The authors appreciate the editor’s comment and the authors have uploaded the minimal data as Supporting information files.

4. Editor: Please review your reference list to ensure that it is complete and correct. If you have cited papers that have been retracted, please include the rationale for doing so in the manuscript text, or remove these references and replace them with relevant current references. Any changes to the reference list should be mentioned in the rebuttal letter that accompanies your revised manuscript. If you need to cite a retracted article, indicate the article’s retracted status in the References list and also include a citation and full reference for the retraction notice.

Authors response: The authors appreciate the editor’s comment. We have reviewed the reference list to ensure it is correct and complete. No changes have been made on the reference list.

Reviewer #2: A well written manuscript, but the abstract had no results. The section of the abstract titled "Methods and Results" only had methods. There was then a conclusion without having provided study results. I found no other significant issues requiring corrections. Authors response: The authors acknowledge the omission. We have included the study results in the abstract section.

Reviewer #3:

General comments

Integrase inhibitor-containing regimens should be abbreviated for easier read. A lot of your abbreviations were not given its full meaning at first mention.. (PLWH, NNRTI, INSTI, DTG, NAFLD etc). Ensure this is written in full for the sake of better understanding of concepts.

The authors would also need to be consistent with your abbreviations, "PLWH' should not be interchanged with "PWH"

Authors response: 

(i)The authors make note of the reviewer comment on abbreviation of integrase inhibitor-containing regimen and wish to state that the integrase inhibitor-containing regimen has been abbreviated as (IICR)

(ii) The authors appreciate the reviewer’s comments and acknowledge that some abbreviations were not written in full at first mention and state that this has been done to improve clarity especially for the non-specialist readers. 

Reviewer's comment

1. I suggest the experimental design for administering different diets (especially for phase 2) to the experimental animals be represented in tabular form for easy reference (Line 101-114)

Authors response: The authors note the reviewer’s suggestion and have included a paradigm illustration for easy reference.

2.The reporting of significant changes in parameters weekly, after which a table to show the same information makes it quite bulky to read. I believe a summary stating the point of significance and the number of weeks for which it was significant should suffice. While the Table can show the full information.

Authors response: The authors have taken note of the reviewer’s comment and have attempted to implement the suggestion. However, reporting of significant changes in parameter on weekly basis in the opinion of the authors improve clarity. Hence, the authors politely request to let the reporting of results to stay as it is. 

3.“On”? Was the FBG recorded on the Call® 123 EZ II or it was an instrument/equipment used to record? If the latter is the case, then “via” should be used (Line 122)

Authors reply: The authors take note of the reviewer’s comments and wish to clarify that (On Call® EZ II) is the trade name of the instrument (glucometer) that was used for determining the fasting blood glucose. 

4.The AUC- What curve is being referred to? Curve of which graph? Where was it mentioned that a graph was plotted or was it stated in the cited reference. (line 131-132).

Authors reply: The authors acknowledge the reviewer’s comment. The graphs been referred to are figure 3 and figure 6.

5.The calculation (Line 139-140) looks like a screenshot cropped and pasted in the document. I suggest the calculation be re-input manually.

The authors appreciate and accept the reviewer’s comment and the formulas have been redone manually 

6.Generally, the use of abbreviations needs to be worked on. The essence of using abbreviations is to help simplify continuous use of some compound terms, for clarity sake. It is noticed that some abbreviations are not defined at first mention (e.g PWLH), some are not used consistently (PWLH/PWH), and some have been previously mentioned but still get defined and mentioned concurrently (e.g AUC. see line 351 for a term that has been defined previously see line 131).

Author response: The authors appreciate the reviewer’s comments and acknowledge that some abbreviations were not written in full at first mention and some previously mentioned still get defined concurrently, this correction has been done to improve clarity.

Discussion: line 1010, I believe the population of sub-saharan Africa is more than 6% of the world's population, ensure your findings are made correctly. 7.Please fact check and add reference.

Author response: The authors admit to the oversight and have added the correct population percentage of sub-saharan which is 18% of the world’s population.

---

## [Decision Letter · Decision Letter 1]

30 Jan 2024

Dietary regimens appear to possess significant effects on the development of Combined Antiretroviral Therapy (cART)-associated metabolic syndrome

PONE-D-23-33547R1

Dear Dr. CHEGE,

We’re pleased to inform you that your manuscript has been judged scientifically suitable for publication and will be formally accepted for publication once it meets all outstanding technical requirements.

Kind regards,

Chika Kingsley Onwuamah, Ph.D.

Academic Editor

PLOS ONE

Additional Editor Comments (optional):

Reviewers' comments:

Reviewer's Responses to Questions

**Comments to the Author**

1. If the authors have adequately addressed your comments raised in a previous round of review and you feel that this manuscript is now acceptable for publication, you may indicate that here to bypass the “Comments to the Author” section, enter your conflict of interest statement in the “Confidential to Editor” section, and submit your "Accept" recommendation.

Reviewer #3: All comments have been addressed

2. Is the manuscript technically sound, and do the data support the conclusions?

Reviewer #3: Yes

3. Has the statistical analysis been performed appropriately and rigorously? 

Reviewer #3: Yes

4. Have the authors made all data underlying the findings in their manuscript fully available?

Reviewer #3: Yes

5. Is the manuscript presented in an intelligible fashion and written in standard English?

Reviewer #3: Yes

6. Review Comments to the Author

Reviewer #3: This manuscript speaks on a study carried out to show the role/effect of dietary regimes in antiretroviral therapy leading to cART-associated metabolic syndrome and the potential of dietary regimens to increase chances of combined antiretroviral therapy-associated metabolic syndrome, bringing to the fore, a need for dietary considerations in drug administration

The authors have addressed the concerns earlier raised, hence no further comments

7. PLOS authors have the option to publish the peer review history of their article (what does this mean?). If published, this will include your full peer review and any attached files.

Reviewer #3: **Yes: **Olufemi Samuel AMOO

---

## [Editor Report · Acceptance letter]

17 Feb 2024

PONE-D-23-33547R1 

PLOS ONE

Dear Dr. CHEGE, 

I'm pleased to inform you that your manuscript has been deemed suitable for publication in PLOS ONE. Congratulations! Your manuscript is now being handed over to our production team.

Kind regards, 

on behalf of

Dr. Chika Kingsley Onwuamah 

Academic Editor

PLOS ONE